# FedARC: Anchor-Guided Residual Compensation for Data and Model Heterogeneous Federated Learning

**Chentao Lu** [1]  **Xuhao Ren** [2]  **Dawei Xu** [1]  **Chuan Zhang** [2]  **Liehuang Zhu** [2]

## Abstract

Federated learning (FL) allows clients to collaboratively train models without exposing private data, but practical FL is simultaneously challenged by data heterogeneity and model heterogeneity. Prior heterogeneous FL (HtFL) approaches often fail to handle fine-grained feature shifts, leading to weak representation alignment and limited cross-client knowledge transfer, which degrades both personalization and generalization. We propose FedARC, an HtFL framework that couples a shared lightweight extractor with client-specific fusion: a trainable projector integrates local and global embeddings, while adaptive residual compensation dynamically corrects feature-level mismatches. To further stabilize aggregation, FedARC performs semantic anchor alignment across clients, and we theoretically prove FedARC converges with a non-convex convergence rate $\mathcal{O}(1/T)$. Experiments on five public benchmarks demonstrate that FedARC outperforms nine state-of-the-art HtFL baselines by up to 2.63% in average accuracy, while maintaining efficient communication and computation.

## 1. Introduction

Federated learning (FL) allows multiple clients to train jointly without sharing raw data. In practice, however, classical algorithms such as FedAvg (McMahan et al., 2017) assume a single architecture across clients, while real deployments exhibit mixed data distributions, device capabilities, and model designs. Specifically, client data are often non-IID (Lu et al., 2024), devices range from phones to edge servers with disparate compute/communication budgets (Yi et al., 2022), and organizations maintain proprietary archi-

tectures to meet internal or IP constraints (Shao et al., 2023). These factors jointly make a "one-size-fits-all" global model brittle (Qi et al., 2024).

Rather than enforcing a single global architecture, heterogeneous federated learning (HtFL) (Tan et al., 2022; Zhang et al., 2025) embraces architectural diversity and non-IID data, enabling collaboration via transferable knowledge, and is therefore widely regarded as a realistic and mainstream paradigm in FL. Existing HtFL approaches can be generally categorized into three groups: (i) *Knowledge-distillation (KD)–based* methods transfer logits/representations using public or proxy data or summary statistics (Morafah et al., 2024; Park et al., 2023; Wu et al., 2022; Zhu et al., 2021); they reduce coupling to a shared architecture but often suffer from limited or biased proxies, additional communication or server-side computation, and potential privacy risks. (ii) *Model-split* methods share only a homogeneous part (e.g., a header or an extractor) while keeping heterogeneous parts local (Yi et al., 2023); fixed partitioning can constrain expressiveness and may leak architectural priors (Yang et al., 2024). (iii) *Mutual-learning* methods co-train a large heterogeneous model together with a small homogeneous proxy per client and aggregate only the proxy (Shen et al., 2020; Weng et al., 2025); yet capacity mismatch and insufficient calibration leave knowledge transfer suboptimal and increase compute and communication costs (Yi et al., 2024b). However, a shared failure mode persists across these categories: batch-level distribution shifts and fine-grained feature misalignment are under-addressed, causing misaligned class centers and overlapping clusters that hurt both generalization and personalization. As illustrated in Fig. 1(a,c), features from conventional HtFL scatter and drift across clients, inducing domain bias and unstable aggregation.

To address these limitations, we propose FedARC—Federated learning with Anchor-Guided Residual Compensation—an HtFL framework that fuses local and global representations via a trainable projector and performs dimension-wise residual compensation to correct distribution shifts within and across clients. A semantic-anchor alignment module further reduces inter-client divergence and stabilizes aggregation. As illustrated in Fig. 1,(c), conventional HtFL misaligns

---

[1]Changchun University, Changchun, China. [2]Beijing Institute of Technology, Beijing, China. Correspondence to: Chuan Zhang <chuanz@bit.edu.cn>.

*Proceedings of the $43^{rd}$ International Conference on Machine Learning*, Seoul, South Korea. PMLR 306, 2026. Copyright 2026 by the author(s).

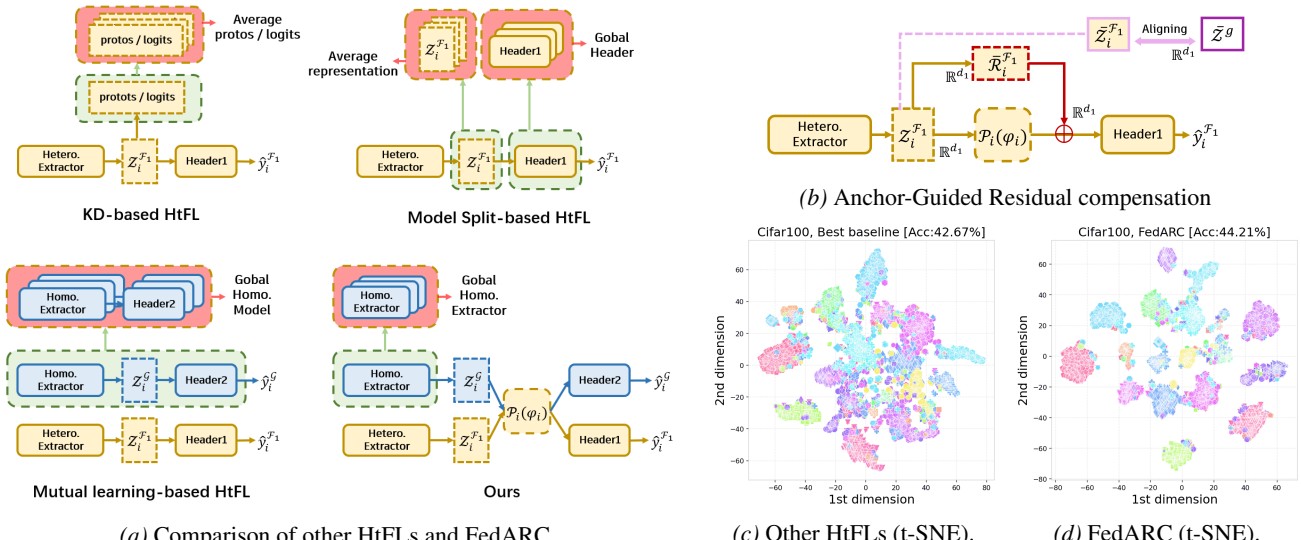

*(a)* Comparison of other HtFLs and FedARC.

*(b)* Anchor-Guided Residual compensation

*(c)* Other HtFLs (t-SNE).

*(d)* FedARC (t-SNE).

*Figure 1.* KD-based HtFL: clients share prediction statistics (logits/prototypes); the server averages and broadcasts guidance for the next round. Model-split HtFL: a unified component (e.g., shared header) is aggregated across clients while the rest stays local. Mutual-learning HtFL: a small homogeneous model is co-trained and aggregated to exchange global knowledge. Ours: only the small homogeneous extractor is uploaded/aggregated; on-client a lightweight projector fuses features and residual compensation (see *(b)*) calibrates the fused representation before either head consumes it, reducing cross-client feature drift. (Red: server-aggregated; green: clients-upload items; yellow: local heterogeneous model/representations module; blue: global homogeneous model/representations module.)

local semantic centers (client means), leading to scattered or overlapping features and domain bias that hampers cross-client transfer. In contrast, FedARC (Fig. 1,(d)) explicitly compensates local representation shifts with residual vectors and anchors each client's overall feature mean to a global semantic anchor, yielding compact, well-separated clusters in a shared semantic space while maintaining global consistency. Consequently, FedARC achieves a balanced trade-off between personalization and generalization by preserving local semantics while aligning to global anchors. The fusion mechanism, combined with residual compensation, adaptively bridges the heterogeneity gap, whereas semantic-anchor alignment provides a stable cross-client target.

The main contributions of this work are listed as follows:

- We calibrate heterogeneous extractors in a fused representation space using a lightweight projector and client-specific residual biases, avoiding reliance on public proxy data or logit-only transfer. We introduce a global semantic anchor computed from clients' fused space means and align both the fused space and shared prefix subspace, stabilizing aggregation under feature drift.

- We provide a theoretical analysis that FedARC achieves the $\mathcal{O}(1/T)$ non-convex convergence rate under standard smoothness and bounded-variance assumptions.

- Extensive experiments on five benchmark datasets under both statistical and model heterogeneity settings demon-

strate that FedARC outperforms nine state-of-the-art HtFL methods by up to 2.63% in average accuracy, while maintaining communication and computation efficiency.

**Conflict of Interest Disclosure.** The authors declare no financial conflicts of interest.

### 1.1. Related Work

Existing HtFL methods are broadly classified as (i) partially heterogeneous, where clients extract submodels from a unified template via pruning or reconfiguration (e.g., EMO (Wu et al., 2025), FCCL+ (Huang et al., 2023), InCo (Chan et al., 2024), FedSA-LoRA (Guo et al., 2025), DepthFL (Kim et al., 2022)); or (ii) fully heterogeneous, where each client uses a distinct architecture, requiring specialized knowledge fusion strategies, which can be further divided as follows:

**Knowledge distillation (KD)-based HtFL.** Existing KD strategies for HtFL fall into three categories: (i) *Statistical transfer*: Methods such as FD (Jeong et al., 2018), FedProto (Tan et al., 2022), PLADA (Fang et al., 2024), FedTGP (Zhang et al., 2024b), FedHCD (Feng et al., 2024), and FedSA (Zhou et al., 2025) exchange only summary statistics or prototypes, reducing bandwidth but still risking membership inference. In addition, FedLabel (Cho et al., 2023) selectively assimilates local or global predictions for semi-supervised FL but still assumes homogeneous models and works at the logit/pseudo-label level. (ii) *Proxy logit synchronization*: Systems such as FedGD (Zhang et al., 2023a),

FZSL (Sun et al., 2024), FedDTG (Gong et al., 2024), and DFRD (Wang et al., 2023) generate synthetic anchors for alignment, but face extra optimization, mode collapse, and privacy leakage if synthetic data memorize local features. (iii) *Synthetic data distillation*: Approaches like Fed-ET (Cho et al., 2022), FSFL (Huang et al., 2022), KRR-KD (Park et al., 2023), and TAKFL (Morafah et al., 2024) synchronize soft outputs using a proxy dataset, but this incurs high communication and exposes models to privacy attacks (e.g., PLI (Takahashi et al., 2023)).

**Model Split-based HtFL.** These methods decompose each complete local model into a feature extractor and a task-specific head. In FedPAC (Xu et al., 2023), FedRep (Collins et al., 2021), FedAS (Yang et al., 2024), and FedAlt/FedSim (Pillutla et al., 2022), aggregate homogeneous headers while keeping heterogeneous extractors. Others, such as LG-FedAvg (Liang et al., 2020), FedGen (Zhu et al., 2021), FedGH (Yi et al., 2023), and CHFL (Liu et al., 2022), aggregate homogeneous extractors while keeping headers private, while $CD^2$-pFed (Shen et al., 2022) focuses on parameter partitioning rather than cross-architecture alignment. However, partial model sharing may inadvertently reveal architectural priors and constrain task adaptability.

**Mutual learning-based HtFL.** These methods train each client's local heterogeneous model together with a lightweight homogeneous auxiliary model; the auxiliary model enables server-side aggregation while the local model preserves local inductive bias (Xu et al., 2024; Zhang et al., 2024a; Louizos et al., 2024). Representative methods (pFedES (Yi et al., 2025), FedKD (Wu et al., 2022), FedMRL (Yi et al., 2024a), Fed-CO2 (Cai et al., 2023) and Fed-RoD (Chen & Chao, 2022)) verify this idea but roughly double compute/communication. Variants (FedAPEN (Qin et al., 2023), FedSKD (Weng et al., 2025), pFedAFM (Yi et al., 2024b)) improve aggregation flexibility yet transfer low-dimensional or indirect signals, limiting fusion fidelity.

Distinctly, our FedARC framework performs calibration in the projector induced fused feature space and introduces inexpensive client-specific residual biases for per-client feature correction. Crucially, FedARC uses a fused space semantic anchor to align cross-client drift in both the fused space and its shared prefix subspace.

## 2. Preliminaries

FedAvg (McMahan et al., 2017) is the canonical FL algorithm, where a central server coordinates $N$ clients. At the beginning of each communication round, the server randomly selects a fraction $\rho$ of $N$ clients, yielding an active set $\mathcal{S}$ with cardinality $K = \rho \cdot N$. The server broadcasts the current global model, denoted as $\mathcal{F}(\omega)$ ($\mathcal{F}(\cdot)$ is complete model structure and $\omega$ its parameters). Each selected client $k$ trains the received model on its private local dataset $\mathcal{D}_k \sim P_k$ ($\mathcal{D}_k$ obeys distribution $P_k$, datasets across clients are generally non-IID). Using gradient descent, the parameters are updated as $\omega_k \leftarrow \omega - \eta \nabla \ell (\mathcal{F}(\boldsymbol{x}_i; \omega), y_i)$, where $\ell(\cdot, \cdot)$ is the sample-wise loss for $(\boldsymbol{x}_i, y_i) \in \mathcal{D}_k$. After local training, each client uploads the updated model parameters $\omega_k$ to the server. The server refines the global model by weighted aggregation, $\omega = \sum_{k \in S} \frac{n_k}{n} \omega_k$ ($n_k = |\mathcal{D}_k|$ is the number of data samples on client $k$, $n = \sum_{k=0}^{N-1} n_k$ is the number of total data samples on all clients).

FedAvg assumes identical model architectures and minimizes the average loss of the global model,

$$\min_{\omega \in \mathbb{R}^d} \sum_{k=0}^{N-1} \frac{n_k}{n} \mathcal{L}_k \Big( \mathcal{F}(\omega); \mathcal{D}_k \Big), \tag{1}$$

where $d$ is the parameter dimension of $\omega$, and $\mathcal{L}_k$ is the average loss of the global model on private data $\mathcal{D}_k$.

In this work, all clients tackle the same prediction task, yet each maintains its own distinct local model with different architecture, $\mathcal{F}_k(\cdot)$ with personalized model parameters $\omega_k$. Our FedARC framework optimizes to minimize the cumulative loss of these personalized models on the private data:

$$\min_{\omega_k \in \mathbb{R}^{d_k}} \sum_{k=0}^{N-1} \mathcal{L}_k \Big( \mathcal{F}_k(\omega_k); \mathcal{D}_k \Big), \tag{2}$$

where the dimensionality $d_k$ varies across all clients.

### 2.1. Overview

Each communication round $t$ in FedARC is as follows:

1. Randomly sampling $K$ clients out of $N$ total participants, the server broadcasts the global feature extractor $\mathcal{G}^{ex}(\theta^{ex,t-1})$ to all selected clients in $\mathcal{S}^t$.

2. Each participating client $k$ jointly trains its local heterogeneous extractor $\mathcal{F}_k^{ex}(\omega_k^{ex,t-1})$ and the received global homogeneous extractor $\mathcal{G}^{ex}(\theta^{ex,t-1})$ on local dataset $(\boldsymbol{x}_i, y_i)$ from $\mathcal{D}_k$. For each batch, the two extractors produce $\mathcal{Z}_i^{\mathcal{F}_k}$ and $\mathcal{Z}_i^{\mathcal{G}}$; a client-specific projector fuses them into $\mathcal{Z}_i$, which is then calibrated by residual biases before being consumed by the heterogeneous headers $\mathcal{F}_k^{hd}(\omega_k^{hd,t-1})$ and homogeneous headers $\mathcal{G}^{hd}(\theta^{hd,t-1})$.

3. Once local training concludes, only the updated homogeneous feature extractor $\mathcal{G}^{ex}(\theta_k^{ex,t})$ is uploaded to the server, the server performs a weighted aggregation to obtain the updated global feature extractor $\mathcal{G}^{ex}(\theta^{ex,t})$.

These steps are executed iteratively until the heterogeneous local models $\mathcal{F}_k(\omega_k)$ of all clients converge. During the inference phase, only the personalized local heterogeneous

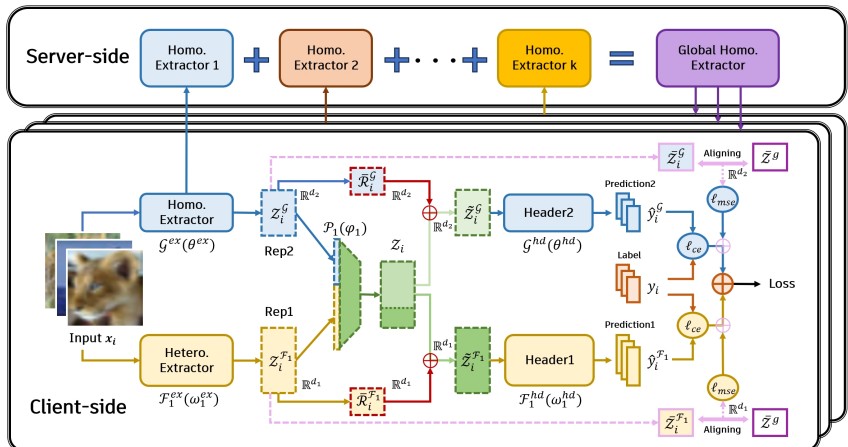

*Figure 2.* Workflow of FedARC. For each client $k$, the heterogeneous extractor $\mathcal{F}_k^{ex}$ ($\omega_k^{ex}$) and the shared homogeneous extractor $\mathcal{G}^{ex}(\theta^{ex})$ encode the same input $x_i$ into local and global features $\mathcal{Z}_i^{\mathcal{F}_k}$ and $\mathcal{Z}_i^{\mathcal{G}}$. A lightweight projector $\mathcal{P}_k(\varphi_k)$ fuses these features into $\mathcal{Z}_i$, which is further adjusted by residual vectors $\bar{\mathcal{R}}_i^{\mathcal{F}_k}$ and $\bar{\mathcal{R}}_i^{\mathcal{G}}$ and anchored to the global mean $\widetilde{\mathcal{Z}}^g$ to correct client-specific distribution shifts. The compensated representations are consumed by the heterogeneous head $\mathcal{F}_k^{hd}$ and homogeneous head $\mathcal{G}^{hd}$ to compute local objectives, while only the parameters of $\mathcal{G}^{ex}$ (blue) are uploaded and aggregated on the server; and local heterogeneous model (yellow) remain on-device.

model of each client is adopted. Further details are provided in Algorithm 1 (refer to Appendix A.2.).

Building on Eq.(2), the training objective is reformulated as minimizing the sum of the loss of the combined model $\mathcal{C}_k(\varepsilon_k) = \mathcal{F}_k(\omega_k) \circ \mathcal{G}(\theta)$ across all clients:

$$\min_{\omega_0, \ldots, \omega_{N-1}, \theta} \sum_{k=0}^{N-1} \mathcal{L}_k \Big( \mathcal{C}_k(\omega_k \circ \theta); \mathcal{D}_k \Big). \quad (3)$$

### 2.2. Feature Representation Fusion

**Motivation.** A straightforward approach is to feed global and local feature representations into their respective headers and jointly optimize them by minimizing the sum of their losses. However, this suffers from two issues: (1) Low-quality representations from the global extractor might exacerbate misalignment with local features; (2) The absence of feature calibration allows local representation bias to accumulate over time, hindering effective knowledge transfer and generalization. To address this, FedARC employs adaptive residual compensation and semantic anchor alignment for fine-grained alignment and fusion of local and global knowledge at the feature level.

### 2.3. Adaptive Residual Compensation

Given a training sample $(\boldsymbol{x}_i, y_i) \in \mathcal{D}_k$, $\boldsymbol{x}_i$ is processed by the heterogeneous extractor $\mathcal{F}_k^{ex}(\omega_k^{ex,t-1})$ and the global homogeneous extractor $\mathcal{G}^{ex}(\theta_k^{ex,t-1})$ to obtain two respective representations,

$$\mathcal{Z}_i^{\mathcal{F}_k} = \mathcal{F}_k^{ex}(\boldsymbol{x}_i; \omega_k^{ex,t-1}), \quad \mathcal{Z}_i^{\mathcal{G}} = \mathcal{G}^{ex}(\boldsymbol{x}_i; \theta^{ex,t-1}), \quad (4)$$

where $\mathcal{Z}_i^{\mathcal{F}_k} \in \mathbb{R}^{d_1}$ contains client-specific local knowledge matching its own data distribution, $\mathcal{Z}_i^{\mathcal{G}} \in \mathbb{R}^{d_2}$ contains generalized knowledge beneficial across all clients. Inspired by widely used knowledge fusion methods (Yi et al., 2024a;b), we concatenate local and global feature representations and project them via a learnable projector $\mathcal{P}_k(\varphi_k^{t-1})$:

$$\mathcal{Z}_i = \mathcal{P}_k\big(\mathcal{Z}_i^{\mathcal{F}_k} \circ \mathcal{Z}_i^{\mathcal{G}}; \varphi_k^{t-1}\big), \quad \mathcal{P}_k : \mathbb{R}^{d_1+d_2} \to \mathbb{R}^{d_1} \quad (5)$$

Due to significant heterogeneity in both data and model architectures, direct feature concatenation may still leave latent semantic discrepancies. To explicitly mitigate this, we consider client-specific learnable calibration vectors (residual biases) $\bar{\mathcal{R}}_i^{\mathcal{F}_k} \in \mathbb{R}^{d_1}$ and $\bar{\mathcal{R}}_i^{\mathcal{G}} \in \mathbb{R}^{d_2}$, learned during training to adjust each client's representation space:

$$\widetilde{\mathcal{Z}}_i^{\mathcal{F}_k} = \mathcal{Z}_i + \bar{\mathcal{R}}_i^{\mathcal{F}_k}, \quad \widetilde{\mathcal{Z}}_i^{\mathcal{G}} = \mathcal{Z}_i^{1:d_2} + \bar{\mathcal{R}}_i^{\mathcal{G}}. \quad (6)$$

The local heterogeneous headers take the full fused $\mathcal{Z}_i \in \mathbb{R}^{d_1}$, while the homogeneous headers operate on its prefix subspace $\mathcal{Z}_i^{1:d_2} \in \mathbb{R}^{d_2}$ with $d_1 > d_2$. This nested slicing preserves a shared subspace for global alignment without disjointly splitting features. These residual compensations are jointly optimized with model parameters, enabling each client to flexibly align fused representations with their unique local semantic spaces $\mathbb{R}^{d_1}, \mathbb{R}^{d_2}$. Each client's heterogeneous prediction header $\mathcal{F}_k^{hd}(\omega_k^{hd})$ and the global homogeneous prediction header $\mathcal{G}^{hd}(\theta^{hd})$ output logits in $\mathbb{R}^L$ ($L$ is the label dimension):

$$\hat{y}_i^{\mathcal{F}_k} = \mathcal{F}_k^{hd}(\widetilde{\mathcal{Z}}_i^{\mathcal{F}_k}; \omega_k^{hd,t-1}), \hat{y}_i^{\mathcal{G}} = \mathcal{G}^{hd}(\widetilde{\mathcal{Z}}_i^{\mathcal{G}}; \theta^{hd,t-1}). \quad (7)$$

The prediction losses (cross-entropy (Zhang & Sabuncu,

2018)) are computed separately for each branch:

$$\ell_i^{\mathcal{F}_k} = \ell_{ce}(\hat{y}_i^{\mathcal{F}_k}, y_i), \quad \ell_i^{\mathcal{G}} = \ell_{ce}(\hat{y}_i^{\mathcal{G}}, y_i). \quad (8)$$

## 2.4. Semantic Anchor Alignment

To mitigate semantic drift and further separate $\mathcal{Z}_i$ and $\bar{\mathcal{R}}_i$, we propose a semantic anchor alignment that encourages the learnable projector to produce $\mathcal{Z}_i$ by a more client-invariant mean, while the residual vectors $\bar{\mathcal{R}}_i^{\mathcal{F}_k}$ remain client-specific offsets that are not directly constrained by this anchor.

Concretely, we align the means $\bar{\mathcal{Z}}_i^{\mathcal{F}_k} \in \mathbb{R}^{d_1}$ and $\bar{\mathcal{Z}}_i^{\mathcal{G}} \in \mathbb{R}^{d_2}$ by aligning them with a global consensus mean $\bar{\mathcal{Z}}^g$ for each feature dimension independently. The global mean $\bar{\mathcal{Z}}^g = \sum_{k=1}^{N} \bar{\mathcal{Z}}_k^g, \forall \bar{\mathcal{Z}}^g \in \mathbb{R}^{d_1}$ is determined during the FL initialization phase. We measure the similarity between each client's mean ($\bar{\mathcal{Z}}_i^{\mathcal{F}_k}$, $\bar{\mathcal{Z}}_i^{\mathcal{G}}$) and the global anchor ($\bar{\mathcal{Z}}^{g,1:d_1} \in \mathbb{R}^{d_1}$, $\bar{\mathcal{Z}}^{g,1:d_2} \in \mathbb{R}^{d_2}$) by the mean squared error (MSE) (Tuchler et al., 2002; Zhang et al., 2023b), with hyperparameter $\lambda$ controlling regularization strength. Empirically, this yields following objective, updating Eq. (8):

$$\begin{aligned} \ell_i^{\mathcal{F}_k} &= \ell_{ce}(\hat{y}_i^{\mathcal{F}_k}, y_i) + \lambda \cdot \ell_{mse}(\bar{\mathcal{Z}}_i^{\mathcal{F}_k}, \bar{\mathcal{Z}}^{g,1:d_1}), \\ \ell_i^{\mathcal{G}} &= \ell_{ce}(\hat{y}_i^{\mathcal{G}}, y_i) + \lambda \cdot \ell_{mse}(\bar{\mathcal{Z}}_i^{\mathcal{G}}, \bar{\mathcal{Z}}^{g,1:d_2}). \end{aligned} \quad (9)$$

In the above, the semantic anchor terms require empirical mean estimates. For each client, we calculate $\bar{\mathcal{Z}}_i^{\mathcal{F}_k} : \hat{\bar{\mathcal{Z}}}_i^{\mathcal{F}_k} = \frac{1}{n_k} \sum_{j=1}^{n_k} \mathcal{F}_k^{ex}(\boldsymbol{x}_{ij}; \omega_k^{ex})$ and $\bar{\mathcal{Z}}_i^{\mathcal{G}} : \hat{\bar{\mathcal{Z}}}_i^{\mathcal{G}} = \frac{1}{n_k} \sum_{j=1}^{n_k} \mathcal{G}^{ex}(\boldsymbol{x}_{ij}; \theta^{ex})$ over the entire local dataset. However, since SGD only accesses mini-batches in each forward pass, we utilize a moving average strategy to approximate these statistics, following (Li et al., 2020; Zhang et al., 2015) we obtain:

$$\begin{aligned} \hat{\bar{\mathcal{Z}}}_i^{\mathcal{F}_k} &= (1-\kappa) \cdot \hat{\bar{\mathcal{Z}}}_i^{\mathcal{F}_k, t-1} + \kappa \cdot \hat{\bar{\mathcal{Z}}}_i^{\mathcal{F}_k, t}, \\ \hat{\bar{\mathcal{Z}}}_i^{\mathcal{G}} &= (1-\kappa) \cdot \hat{\bar{\mathcal{Z}}}_i^{\mathcal{G}, t-1} + \kappa \cdot \hat{\bar{\mathcal{Z}}}_i^{\mathcal{G}, t}, \end{aligned} \quad (10)$$

where $\hat{\bar{\mathcal{Z}}}_i^{\mathcal{F}_k, t-1}$, $\hat{\bar{\mathcal{Z}}}_i^{\mathcal{G}, t-1}$ are from the previous batch, and $\hat{\bar{\mathcal{Z}}}_i^{\mathcal{F}_k, t}$, $\hat{\bar{\mathcal{Z}}}_i^{\mathcal{G}, t}$ are from the current batch, $\kappa$ serves as a momentum coefficient that balances historical and current batch statistics. Since the two feature extractors are updated locally but may be reset between global rounds, we re-initialize the running-mean estimators at the beginning of each round for numerical stability, while keeping the server-side global anchor fixed.

Finally, we compute the total loss by weighting the two branches using coefficients $\alpha_i^{\mathcal{F}_k}$ and $\alpha_i^{\mathcal{G}}$:

$$\ell_i = \alpha_i^{\mathcal{F}_k} \cdot \ell_i^{\mathcal{F}_k} + \alpha_i^{\mathcal{G}} \cdot \ell_i^{\mathcal{G}}. \quad (11)$$

By default, both weights are set to $\alpha_i^{\mathcal{F}_k} = \alpha_i^{\mathcal{G}} = 1$ to ensure equal contribution. The composite loss $\ell_i$ is jointly

optimized for the heterogeneous local model, global feature extractor, and projector using SGD following FedAvg,

$$\begin{aligned} \omega_k^t &\leftarrow \omega_k^{t-1} - \eta_\omega \nabla \ell_i, \\ \theta_k^t &\leftarrow \theta^{t-1} - \eta_\theta \nabla \ell_i, \\ \varphi_k^t &\leftarrow \varphi_k^{t-1} - \eta_\varphi \nabla \ell_i, \end{aligned} \quad (12)$$

where the learning rates $\eta_\omega, \eta_\theta, \eta_\varphi$ are set identically to ensure convergence stability. This unified training procedure encourages representations to be simultaneously generalizable and personalized. In implementation, we update $\bar{\mathcal{R}}_i^{\mathcal{F}_k}$, $\bar{\mathcal{R}}_i^{\mathcal{G}}$ with the same SGD (omitted in Eq. (12) for brevity).

## 3. Convergence Analysis

We denote notations following (Yi et al., 2025; 2024a). Let $t$ denote the communication round, and $e \in \{0, 1, ..., E\}$ the iterations of local training within each round. At the start of round $t + 1$ (iteration $tE + 0$), client $k$ receives the global feature extractor $\mathcal{G}^{ex}(\theta^{ex,t})$ from the server. The $e$-th local update is indexed by $tE + e$, and $tE + E$ indicates the completion of local training in round $t + 1$, after which the client uploads its updated $\mathcal{G}^{ex}(\theta_k^{ex,t+1})$ for aggregation. We modify client $k$'s combined local model as $\mathcal{C}_k(\varepsilon_k) = (\mathcal{F}_k(\omega_k) \circ \mathcal{G}(\theta) | \mathcal{P}_k(\varphi_k))$, where $\mathcal{F}_k(\omega_k)$ is the heterogeneous model, $\mathcal{G}(\theta)$ is the small homogeneous model, and $\mathcal{P}_k(\varphi_k)$ is the learnable projector. The learning rate for each component is $\eta = \{\eta_\omega, \eta_\theta, \eta_\varphi\}$.

**Assumption 3.1. Lipschitz Smoothness.** For each client $k$, the gradients of its combined local heterogeneous model $\varepsilon_k$ satisfy $L_1$–Lipschitz smoothness (Tan et al., 2022),

$$\left\| \nabla \mathcal{L}_k^{t_1}(\varepsilon_k^{t_1}; \boldsymbol{x}, y) - \nabla \mathcal{L}_k^{t_2}(\varepsilon_k^{t_2}; \boldsymbol{x}, y) \right\| \leqslant L_1 \left\| \varepsilon_k^{t_1} - \varepsilon_k^{t_2} \right\|, \quad (13)$$

$$\forall t_1, t_2 > 0, k \in \{0, 1, \dots, N-1\}, (\boldsymbol{x}, y) \in \mathcal{D}_k.$$

The above formulation can be re-expressed as:

$$\mathcal{L}_k^{t_1} - \mathcal{L}_k^{t_2} \leqslant \left\langle \nabla \mathcal{L}_k^{t_2}, \left( \varepsilon_k^{t_1} - \varepsilon_k^{t_2} \right) \right\rangle + \frac{L_1}{2} \left\| \varepsilon_k^{t_1} - \varepsilon_k^{t_2} \right\|_2^2. \quad (14)$$

**Assumption 3.2. Unbiased Gradient and Bounded Variance.** For each client $k$, the stochastic gradient $g_{\varepsilon,k}^t = \nabla \mathcal{L}_k^t(\varepsilon_k^t; \mathcal{B}_k^t)$ computed over mini-batch $\mathcal{B}_k^t$ is unbiased:

$$\mathbb{E}_{\mathcal{B}_k^t \subseteq \mathcal{D}_k}\left[ g_{\varepsilon,k}^t \right] = \nabla \mathcal{L}_k^t\left( \varepsilon_k^t \right), \quad (15)$$

and the variance of $g_{\varepsilon,k}^t$ is bounded by:

$$\mathbb{E}_{\mathcal{B}_k^t \subseteq \mathcal{D}_k}\left[ \left\| \nabla \mathcal{L}_k^t(\varepsilon_k^t; \mathcal{B}_k^t) - \nabla \mathcal{L}_k^t(\varepsilon_k^t) \right\|_2^2 \right] \leqslant \sigma^2. \quad (16)$$

**Assumption 3.3. Bounded Parameter Variation.** Before and after aggregation on FL server, the deviations in parameters of the global feature extractors $\theta_k^t$ and $\theta$ are subject to the following constraints:

$$\left\| \theta^t - \theta_k^t \right\|_2^2 \leq \delta^2. \quad (17)$$

*Table 1.* Average test accuracy (%) on four datasets in cross-silo and cross-device settings under label skew using HtFE$^{img}_8$.

| Settings | Cross-silo (N=10, $\rho$=100%, $\beta$=0.1) | | | | Cross-device (N=50, $\rho$=20%, $\beta$=0.1) | | | |
|---|---|---|---|---|---|---|---|---|
| Datasets | Cifar10 | Cifar100 | Flowers102 | Tiny* | Cifar10 | Cifar100 | Flowers102 | Tiny* |
| FD | 87.78±0.14 | 42.67±0.07 | 53.31±0.65 | 25.95±0.07 | 84.07±0.15 | 39.62±0.09 | 47.82±0.54 | 24.57±0.12 |
| FedProto | 84.04±0.19 | 36.12±0.10 | 40.12±0.18 | 19.31±0.12 | 67.89±0.41 | 20.82±0.07 | 35.08±0.19 | 18.73±0.21 |
| FedTGP | 87.64±0.12 | 40.69±0.19 | 55.57±0.23 | 27.43±0.08 | 80.57±0.24 | 37.34±0.10 | 47.57±0.26 | 25.64±0.17 |
| LG-FedAvg | 86.47±0.10 | 40.16±0.07 | 46.39±0.39 | 26.08±0.06 | 83.07±0.28 | 38.01±0.08 | 41.21±0.42 | 24.56±0.05 |
| FedGen | 85.34±0.23 | 38.68±0.15 | 45.05±0.16 | 20.68±0.08 | 80.89±0.33 | 36.18±0.12 | 41.27±0.21 | 18.86±0.13 |
| FedGH | 86.07±0.16 | 41.61±0.08 | 47.31±0.15 | 25.41±0.12 | 82.91±0.23 | 37.63±0.09 | 42.85±0.21 | 25.18±0.15 |
| pFedES | 87.54±0.28 | 42.37±0.26 | 51.42±0.26 | 26.87±0.31 | 83.64±0.32 | 38.77±0.35 | 47.77±0.46 | 25.33±0.33 |
| FedKD | 87.64±0.14 | 41.91±0.21 | 50.48±0.25 | 26.26±0.16 | 83.16±0.17 | 38.36±0.11 | 44.86±0.23 | 24.93±0.19 |
| FedMRL | 87.20±0.26 | 42.57±0.32 | 51.65±0.21 | 27.37±0.28 | 83.78±0.31 | 38.93±0.33 | 45.65±0.24 | 25.43±0.30 |
| FedARC | **89.22±0.07** | **44.21±0.05** | **57.67±0.11** | **29.91±0.14** | **86.14±0.14** | **42.25±0.07** | **50.18±0.15** | **27.10±0.16** |

Given the above assumptions, we obtain the following results (proofs provided in the Appendix).

**Lemma 3.4.** *Under Assumptions 3.1 and 3.2, for arbitrary client, the loss across local training steps $\{0, 1, \ldots, E\}$ in the $(t + 1)$-th round is bounded by:*

$$\mathbb{E}\left[\mathcal{L}_{(t+1)E}\right] \leq \mathcal{L}_{tE+0} + \left(\frac{L_1\eta^2}{2} - \eta\right)\sum_{e=0}^{E}\|\nabla\mathcal{L}_{tE+e}\|_2^2 + \frac{L_1E\eta^2\sigma^2}{2}.$$

(18)

**Lemma 3.5.** *Given Assumptions 3.2 and 3.3, the loss of any client before and after the aggregation of the lightweight global homogeneous feature extractors on the FL server can be bounded after the $(t + 1)$-th local training round as:*

$$\mathbb{E}\left[\mathcal{L}_{(t+1)E+0}\right] \leq \mathbb{E}\left[\mathcal{L}_{tE+1}\right] + \eta\delta^2.$$

(19)

**Theorem 3.6.** *One Complete Round of FL. Leveraging Lemma 3.4 and Lemma 3.5, we derive the following result for any client after local updates, server aggregation, and obtaining the updated global feature extractor:*

$$\mathbb{E}\left[\mathcal{L}_{(t+1)E+0}\right] \leq \mathcal{L}_{tE+0} + \left(\frac{L_1\eta^2}{2} - \eta\right)\sum_{e=0}^{E}\|\nabla\mathcal{L}_{tE+e}\|_2^2$$
$$+ \frac{L_1E\eta^2\sigma^2}{2} + \eta\delta^2.$$

(20)

**Theorem 3.7.** *Rate of Convergence for FedARC Under Non-Convex. Combining the preceding assumptions and lemmas, the result below holds for all clients and any given constant $\epsilon > 0$:*

$$\frac{1}{T}\sum_{t=0}^{T-1}\sum_{e=0}^{E-1}\|\nabla\mathcal{L}_{tE+e}\|_2^2 \leq \frac{\frac{1}{T}\sum_{t=0}^{T-1}\left(\mathcal{L}_{tE+0} - \mathbb{E}(\mathcal{L}_{(t+1)E+0})\right)}{\eta - \frac{L_1\eta^2}{2}}$$
$$+ \frac{\frac{L_1E\eta^2\sigma^2}{2} + \eta\delta^2}{\eta - \frac{L_1\eta^2}{2}} < \epsilon,$$
$$s.t. \ \eta < \frac{2\left(\epsilon - \delta^2\right)}{L_1\left(\epsilon + E\sigma^2\right)}$$

(21)

Consequently, under the above conditions, each client's local model in FedARC is guaranteed to achieve a non-convex convergence rate of $\epsilon \sim \mathcal{O}(1/T)$, provided that the learning rates for the local heterogeneous model, homogeneous feature extractor, and learnable projector satisfy the specified constraints.

## 4. Experimental Evaluation

**Datasets.** We benchmark our approach across five broadly adopted image classification datasets, including Cifar10/100 (Krizhevsky et al., 2009), Flowers102 (Nilsback & Zisserman, 2008), Tiny-ImageNet (Chrabaszcz et al., 2017), and DomainNet (Peng et al., 2019).

**Statistical/Model heterogeneity.** To simulate statistical heterogeneity, we partition client data using a Dirichlet distribution as in prior works (Li et al., 2021; 2022; Zhang et al., 2025). Specifically, for each class $c$ and client $k$, $q_{c,k} \sim \text{Dir}(\beta)$ (with $\beta = 0.1$ by default) determines the proportion of samples from class $c$ assigned to client $k$. For every client, the local private dataset is divided into a 75% for training and 25% for testing, and evaluation is performed on the test set. To simulate model heterogeneity among clients, we follow FedTGP (Zhang et al., 2024b), where each client is equipped with a heterogeneous model architecture. This setting is denoted as "HtFE$^{img}_X$", with $X$ indicating the number of different feature extractor architectures in the FL. Each client $k$ is allocated the $(k \bmod X)$-th model structure selected from the candidate architecture pool. Specifically, for the "HtFE$^{img}_8$" setting in our main experiments, we employ eight diverse architectures: 4-layer CNN (McMahan et al., 2017), GoogLeNet (Szegedy et al., 2015), MobileNetV2 (Sandler et al., 2018), and ResNet18/34/50/101/152 (He et al., 2016). To ensure feature dimension consistency across architectures, we add the same average pooling layer behind every extractor, resulting in a unified output dimension $d_1$ (set to 512 by default, $d_2$ set to 256). The homogeneous model used for all clients is a 4-layer CNN.

**Implementation details.** We benchmark FedARC against 9 representative state-of-the-art baseline methods, each representing a prominent approach from the three main branches of fully HtFL, as detailed in the Related Work section. Knowledge distillation: FD (Jeong et al., 2018), FedProto (Tan et al., 2022) and FedTGP (Zhang et al., 2024b). Model split: LG-FedAvg (Liang et al., 2020), FedGen (Zhu

*Table 2.* The test accuracy (%) on Cifar100 under the cross-device setting with various model heterogeneity. Δ: The largest accuracy difference among HtFE$^{img}_2$, HtFE$^{img}_3$, HtFE$^{img}_5$ and HtFE$^{img}_9$. Please refer to the appendix for more model details.

| Settings | Heterogeneous Feature Extractors | | | | | Heterogeneous Models | | |
|---|---|---|---|---|---|---|---|---|
| | HtFE$^{img}_2$ | HtFE$^{img}_3$ | HtFE$^{img}_5$ | HtFE$^{img}_9$ | Δ | Res34-HtC$^{img}_4$ | HtFE$^{img}_8$-HtC$^{img}_4$ | HtM$^{img}_{10}$ |
| FD | 42.88±0.15 | 40.92±0.36 | 39.98±0.13 | 36.62±0.31 | 6.26 | 41.27±0.15 | 38.76±0.08 | 39.82±0.05 |
| FedProto | 35.63±0.22 | 29.89±0.48 | 27.75±0.64 | 21.91±0.28 | 13.72 | 28.62±0.21 | 21.75±0.71 | 32.26±0.13 |
| FedTGP | 42.14±0.29 | 37.97±0.57 | 37.66±0.42 | 36.84±0.61 | **5.30** | 42.21±0.36 | 39.35±0.17 | 38.71±0.18 |
| LG-FedAvg | 41.22±0.13 | 40.59±0.42 | 39.66±0.23 | 35.39±0.11 | 5.83 | - | - | - |
| FedGen | 40.05±0.54 | 38.47±0.29 | 38.48±0.28 | 34.67±0.12 | 5.38 | - | - | - |
| FedGH | 40.11±0.18 | 39.82±0.21 | 38.14±0.18 | 34.38±0.31 | 5.73 | - | - | - |
| pFedES | 42.16±0.12 | 40.69±0.57 | 38.93±0.31 | 36.47±0.48 | 5.69 | 41.07±0.38 | 38.96±0.27 | **40.04±0.25** |
| FedKD | 42.19±0.09 | 40.53±0.07 | 37.47±0.21 | 35.82±0.25 | 6.37 | 37.57±0.45 | 37.89±0.48 | 37.63±0.13 |
| FedMRL | 42.27±0.09 | 40.76±0.61 | 39.42±0.24 | 36.55±0.42 | 5.72 | 41.21±0.42 | 39.58±0.25 | 39.73±0.21 |
| FedARC | **44.36±0.12** | **42.72±0.08** | **41.28±0.06** | **37.28±0.18** | 7.08 | **42.91±0.09** | **40.39±0.13** | 39.74±0.10 |

et al., 2021) and FedGH (Yi et al., 2023). Mutual learning: pFedES (Yi et al., 2025), FedKD (Wu et al., 2022) and FedMRL (Yi et al., 2024a). We adopt two prevalent FL settings in our evaluation, namely cross-silo and the cross-device setting (Kairouz & McMahan, 2021). For the cross-silo setting, we employ a federation of 10 clients, all of which participate in each communication round (i.e., participation ratio $\rho = 1$). For the cross-device setting, we set the total number of clients $N$ to 50, with a random subset of 20% (i.e., $\rho = 0.2$) selected in each round. The cross-device setting serves as the primary focus of our study, as it closely mirrors practical federated deployments involving heterogeneous, resource-constrained edge devices. Without additional specification, we adopt previous work (Zhang et al., 2024b): in each round, each selected client carries out one local training epoch, utilizing a mini-batch size of 10 and a unified learning rate of $\eta_\omega = \eta_\theta = \eta_\varphi = 0.01$. All experiments run for 1,000 communication rounds and are repeated three times.

### 4.1. Comparison Results

**Average Accuracy.** As shown in Table 1, FedARC achieves the best average accuracy across all benchmarks, outperforming 9 state-of-the-art HtFL baselines by up to 2.63% in cross-silo and cross-device settings. By aligning auxiliary and local embeddings in a shared subspace and regularizing class means, FedARC corrects feature misalignment that logit-only distillation (e.g., FD, FedKD) overlooks under data and model heterogeneity. Compared with prototype methods (e.g., FedProto, FedTGP), its anchor-based first-order alignment avoids enforcing a single global prototype, adapts to client drift, and improves accuracy and convergence. Further experiments focus on more challenging Cifar10/100 datasets to rigorously evaluate HtFL performance.

**Impact of Model Heterogeneity.** As shown in Table 2, FedARC achieves the best performance in the cross-device settings under both data and model heterogeneity. Compared to FD, which distills from averaged global logits with-

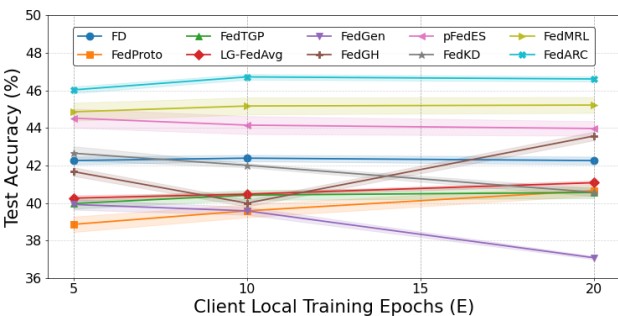

*Figure 3.* Test accuracy (%) under the cross-silo setting on Cifar100 using HtFE$^{img}_8$ with various local training epochs $E$.

*Table 3.* The key component ablation analysis of FedARC on Cifar100 using HtFE$^{img}_8$. FRF: feature representation fusion. ARC: adaptive residual compensation. SAA: semantic anchor alignment.

| FRF | ARC | SAA | Acc(%) |
|---|---|---|---|
| ✓ | | | 40.13 |
| ✓ | ✓ | | 43.64 (↑ **3.51**) |
| ✓ | | ✓ | 41.39 (↑ **1.26**) |
| ✓ | ✓ | ✓ | **44.21** (↑ **4.08**) |

out addressing feature misalignment, FedARC leverages projection-based fusion and residual compensation to better align semantics across clients. FedTGP improves separability via prototypes, but its fixed anchors adapt poorly as heterogeneity increases. FedGH is efficient but the backbone–head decoupling hurts generalization. The best mutual learning baselines (e.g., FedMRL, pFedES) align predictions but not structure, risking representation collapse. In contrast, FedARC explicitly aligns heterogeneous representations, enabling more stable and transferable learning under both data and model heterogeneity.

**Impact of Local Training Epochs.** Increasing the number of client-side local epochs ($E$) can reduce the number of communication rounds in FL. As shown in Fig. 3, mutual learning methods (e.g., pFedES and FedKD) degrade as $E$

*Table 4.* Test accuracy (%) on Cifar100 using HtFE$^{img}{}_8$ with different $\lambda$ and $\kappa$ (default: $\lambda = 1$, $\kappa = 0.1$).

| | $\kappa = 0.1$ | | | | $\lambda = 1$ | | | |
| --- | --- | --- | --- | --- | --- | --- | --- | --- |
| | $\lambda = 0.1$ | $\lambda = 1$ | $\lambda = 5$ | $\lambda = 10$ | $\kappa = 0.05$ | $\kappa = 0.1$ | $\kappa = 0.5$ | $\kappa = 1$ |
| Acc. | 42.92±0.13 | **44.21±0.05** | 43.14±0.37 | 42.46±0.81 | 43.65±0.10 | **44.21±0.05** | 41.87±0.08 | 40.46±0.15 |

*Table 5.* Communication and computation costs on Cifar100 using HtFE$^{img}{}_8$. MB (megabytes), s (seconds).

| Items | Comm. (MB) | | Computation (s) | |
| --- | --- | --- | --- | --- |
| | Up. | Down. | Client | Server |
| FD | 0.52 | 0.89 | 6.54 | 0.04 |
| FedProto | 3.17 | 5.01 | 6.68 | 0.05 |
| FedTGP | 3.17 | 5.01 | 6.61 | 7.91 |
| LG-FedAvg | 5.81 | 5.81 | 6.22 | 0.05 |
| FedGen | 5.81 | 30.22 | 5.79 | 3.02 |
| FedGH | 3.17 | 4.81 | 9.75 | 0.41 |
| pFedES | 39.87 | 39.87 | 17.63 | 0.07 |
| FedKD | 43.24 | 43.24 | 8.14 | 0.08 |
| FedMRL | 50.75 | 50.75 | 8.27 | 0.08 |
| FedARC | 39.87 | 39.87 | 7.83 | 0.08 |

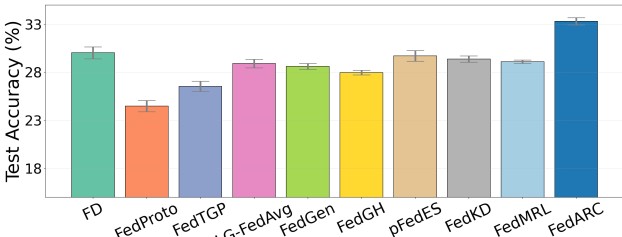

*Figure 4.* Test accuracy (%) under the feature shift scenario on DomainNet using HtFE$^{img}{}_5$.

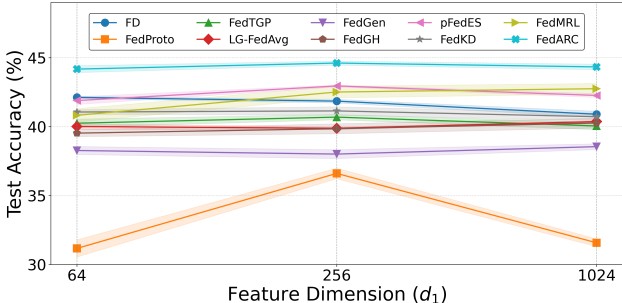

*Figure 5.* Test accuracy (%) under the Dirichlet scenario on Cifar100 using HtFE$^{img}{}_8$ with varying feature dimensions $d_1$.

grows. We attribute this to their reliance on an auxiliary model: longer local training amplifies client-specific bias in the auxiliary branch, which then propagates during aggregation. In contrast, FedMRL and FedARC mitigate this by fusing features from the auxiliary and local models. Moreover, FedARC leverages residual compensation and anchor alignment to regularize client features, yielding the highest accuracy across all $E$.

**Ablation and Hyperparameter Analysis.** Table 3 shows ablation results on the key modules of FedARC. ARC on the top of FRF brings a significant accuracy gain (+3.51%), while SAA further improves performance (+1.26%). Combining all components achieves the best result (44.21%), highlighting complementary roles of ARC and SAA. Table 4 analyzes hyperparameter sensitivity under cross-silo setting. The optimal accuracy is reached with $\lambda = 1$ and $\kappa = 0.1$; either too large or too small values degrade performance, demonstrating that FedARC is robust across a broad hyperparameter range and consistently outperforms other HtFL methods even in less optimal settings.

**Communication and Computation Costs.** We measure communication overhead as the total upload and download bytes per round using the float32 data type in PyTorch (Paszke et al., 2019), and computation as the average GPU time for each client and server on idle GPUs. From Table 5, we observe: (1) Mutual learning methods, despite transmitting smaller models, still incur high communication, and SVD in FedKD does not bring notable savings. (2) KD-based methods are communication-efficient but limited by the lower information capacity of prototypes/logits, resulting in lower accuracy. (3) FedGen and FedTGP involve extra server-side training and multiple rounds, leading

to higher computational power consumption on the server than other HtFLs. Overall, FedARC offers the best trade-off—highest accuracy with no higher communication than other mutual learning methods.

**Performance in the Feature Shift.** From Fig. 4, FedARC delivers superior results, surpassing FD and LG-FedAvg. Prototype-sharing methods degrade notably under cross-domain features, while mutual learning baselines cluster in a similar mid-range. FedProto shows a significant performance gap, as large cross-domain style gaps cause feature means to diverge, making the global prototypes biased and prone to drift without explicit alignment.

**Impact of Feature Dimensions.** Fig. 5 shows that accuracy generally increases steadily with the feature dimension $d_1$ ranging from 64 to 256. Partial parameter–sharing methods (LG-FedAvg, FedGen) deviate from this trend. Most methods peak at $d_1 = 256$, whereas FedMRL and FedGH continue to rise at $d_1 = 1024$, while others (e.g., FD, FedProto) plateau or decline, suggesting that excessively large feature spaces can complicate optimization or invite overfitting—especially for standard aggregation or KD-based methods. Notably, FedARC remains best across all $d_1$.

# 5. Conclusion

In this paper, we introduce a novel HtFL approach, FedARC, leveraging shared homogeneous feature extractors for effective privacy protection and lower communication and computational overhead. The framework allows each client to alternately optimize homogeneous feature extractors and heterogeneous local models, facilitating global and local knowledge exchange. By aggregating the homogeneous local feature extractors from all clients, knowledge across heterogeneous participants can be effectively fused. Both theoretical analysis and experiments verify its effectiveness and efficiency in communication and computation.

## Acknowledgements

This work was supported in part by the National Natural Science Foundation of China under Grant 62472032, in part by the Young Elite Scientists Sponsorship Program by CAST under Grant 2023QNRC001, and in part by the Key Laboratory of Computing Power Network and Information Security, Ministry of Education under Grant 2024PY020.

## Impact Statement

This work studies heterogeneous federated learning under simultaneous data and model heterogeneity and proposes FedARC to improve cross-client representation alignment. By improving robustness across heterogeneous devices and institutions, the method may broaden accessibility of collaborative training, but it can also inherit or amplify biases present in client data; practitioners should evaluate performance across subpopulations and deployment domains. Apart from these considerations, we do not identify additional significant societal impacts that require specific attention.

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

# A. Appendix: Supplementary Material

## A.1. THEORETICAL PROOFS

### A.1.1. PROOF FOR LEMMA 1

An arbitrary client $k$'s local mixed complete model $\varepsilon_t = \varepsilon_k^t \equiv \mathcal{C}_k(\varepsilon_k^t)$ can be updated by $\varepsilon_{t+1} = \varepsilon_t - \eta g_{\varepsilon,t}$ in the $(t+1)$-th round, and following Assumption 1, we can obtain

$$
\begin{aligned}
\mathcal{L}_{tE+1} \leq & \mathcal{L}_{tE+0} + \langle \nabla \mathcal{L}_{tE+0}, (\varepsilon_{tE+1} - \varepsilon_{tE+0}) \rangle + \frac{L_1}{2} \| \varepsilon_{tE+1} - \varepsilon_{tE+0} \|_2^2 \\
= & \mathcal{L}_{tE+0} - \eta \langle \nabla \mathcal{L}_{tE+0}, g_{\varepsilon,tE+0} \rangle + \frac{L_1 \eta^2}{2} \| g_{\varepsilon,tE+0} \|_2^2 .
\end{aligned}
\tag{22}
$$

Taking the expectation of both sides of the inequality concerning the random variable $\xi_{tE+0}$, we obtain

$$
\begin{aligned}
\mathbb{E}\left[\mathcal{L}_{tE+1}\right] \leq & \mathcal{L}_{tE+0} - \eta \mathbb{E}\left[\langle \nabla \mathcal{L}_{tE+0}, g_{\varepsilon,tE+0} \rangle\right] + \frac{L_1 \eta^2}{2} \mathbb{E}\left[\| g_{\varepsilon,tE+0} \|_2^2\right] \\
\overset{(a)}{=} & \mathcal{L}_{tE+0} - \eta \| \nabla \mathcal{L}_{tE+0} \|_2^2 + \frac{L_1 \eta^2}{2} \mathbb{E}\left[\| g_{\varepsilon,tE+0} \|_2^2\right] \\
\overset{(b)}{\leq} & \mathcal{L}_{tE+0} - \eta \| \nabla \mathcal{L}_{tE+0} \|_2^2 + \frac{L_1 \eta^2}{2} \left( \mathbb{E}\left[\| g_{\varepsilon,tE+0} \|\right]_2^2 + \mathcal{V}\left(g_{\varepsilon,tE+0}\right) \right) \\
\overset{(c)}{=} & \mathcal{L}_{tE+0} - \eta \| \nabla \mathcal{L}_{tE+0} \|_2^2 + \frac{L_1 \eta^2}{2} \left( \| \nabla \mathcal{L}_{tE+0} \|_2^2 + \mathcal{V}\left(g_{\varepsilon,tE+0}\right) \right) \\
\overset{(d)}{\leq} & \mathcal{L}_{tE+0} - \eta \| \nabla \mathcal{L}_{tE+0} \|_2^2 + \frac{L_1 \eta^2}{2} \left( \| \nabla \mathcal{L}_{tE+0} \|_2^2 + \sigma^2 \right) \\
= & \mathcal{L}_{tE+0} + \left( \frac{L_1 \eta^2}{2} - \eta \right) \| \nabla \mathcal{L}_{tE+0} \|_2^2 + \frac{L_1 \eta^2 \sigma^2}{2} .
\end{aligned}
\tag{23}
$$

(a), (c), (d) follow Assumption 2 and (b) follows $\mathcal{V}(x) = \mathbb{E}\left[x^2\right] - \left(\mathbb{E}[x]^2\right)$.

Taking the expectation of both sides of the inequality for the model $\varepsilon$ over $E$ iterations, we obtain

$$
\mathbb{E}\left[\mathcal{L}_{tE+1}\right] \leq \mathcal{L}_{tE+0} + \left( \frac{L_1 \eta^2}{2} - \eta \right) \sum_{e=1}^{E} \| \nabla \mathcal{L}_{tE+e} \|_2^2 + \frac{L_1 E \eta^2 \sigma^2}{2} .
\tag{24}
$$

### A.1.2. PROOF FOR LEMMA 2

$$
\begin{aligned}
\mathcal{L}_{(t+1)E+0} = & \mathcal{L}_{(t+1)E} + \mathcal{L}_{(t+1)E+0} - \mathcal{L}_{(t+1)E} \\
\overset{(a)}{\approx} & \mathcal{L}_{(t+1)E} + \eta \left\| \theta_{(t+1)E+0} - \theta_{(t+1)E} \right\|_2^2 \\
\overset{(b)}{\leq} & \mathcal{L}_{(t+1)E} + \eta \delta^2
\end{aligned}
\tag{25}
$$

(a): we can use the gradient of parameter variations to approximate the loss variations, i.e., $\Delta \mathcal{L} \approx \eta \cdot \| \Delta \theta \|_2^2$. (b) follows Assumption 3. Taking the expectation of both sides of the inequality to the random variable $\tau$, we obtain

$$
\mathbb{E}\left[\mathcal{L}_{(t+1)E+0}\right] \leq \mathbb{E}\left[\mathcal{L}_{tE+1}\right] + \eta \delta^2 .
\tag{26}
$$

### A.1.3. PROOF FOR THEOREM 1

Substituting Lemma 1 into the right side of Eq. (24)'s inequality, we obtain

$$
\mathbb{E}\left[\mathcal{L}_{(t+1)E+0}\right] \leq \mathcal{L}_{tE+0} + \left( \frac{L_1 \eta^2}{2} - \eta \right) \sum_{e=0}^{E} \| \nabla \mathcal{L}_{tE+e} \|_2^2 + \frac{L_1 E \eta^2 \sigma^2}{2} + \eta \delta^2 .
\tag{27}
$$

### A.1.4. PROOF FOR THEOREM 2

Interchanging the left and right sides of Eq. (27), we obtain

$$\sum_{e=0}^{E}\|\nabla\mathcal{L}_{tE+e}\|_2^2 \leq \frac{\mathcal{L}_{tE+0} - \mathbb{E}\left[\mathcal{L}_{(t+1)E+0}\right] + \frac{L_1 E \eta^2 \sigma^2}{2} + \eta\delta^2}{\eta - \frac{L_1\eta^2}{2}}. \tag{28}$$

Taking the expectation of both sides of the inequality over rounds $t = [0, T-1]$, we obtain

$$\frac{1}{T}\sum_{t=0}^{T-1}\sum_{e=0}^{E-1}\|\nabla\mathcal{L}_{tE+e}\|_2^2 \leq \frac{\frac{1}{T}\sum_{t=0}^{T-1}[\mathcal{L}_{tE+0} - \mathbb{E}\left[\mathcal{L}_{(t+1)E+0}\right]] + \frac{L_1 E \eta^2 \sigma^2}{2} + \eta\delta^2}{\eta - \frac{L_1\eta^2}{2}}. \tag{29}$$

Let $\Delta = \mathcal{L}_{t=0} - \mathcal{L}^* > 0$, then
$\sum_{t=0}^{T-1}\left[\mathcal{L}_{tE+0} - \mathbb{E}\left[\mathcal{L}_{(t+1)E+0}\right]\right] \leq \Delta$, we can get

$$\frac{1}{T}\sum_{t=0}^{T-1}\sum_{e=0}^{E-1}\|\nabla\mathcal{L}_{tE+e}\|_2^2 \leq \frac{\frac{\Delta}{T} + \frac{L_1 E \eta^2 \sigma^2}{2} + \eta\delta^2}{\eta - \frac{L_1\eta^2}{2}}. \tag{30}$$

If the above equation can converge to a constant $\epsilon$, i.e.,

$$\frac{1}{T}\sum_{t=0}^{T-1}\sum_{e=0}^{E-1}\|\nabla\mathcal{L}_{tE+e}\|_2^2 \leq \frac{\frac{\Delta}{T} + \frac{L_1\eta^2\left(\sigma^2+\delta^2\right)}{2}}{\eta\tilde{\mu} - \frac{L_1\eta^2\tilde{\mu}^2}{2}} < \epsilon, \tag{31}$$

then

$$T > \frac{\Delta}{\epsilon\left(\eta - \frac{L_1\eta^2}{2}\right) - \frac{L_1 E \eta^2 \sigma^2}{2} - \eta\delta^2}. \tag{32}$$

Since $T > 0$, $\Delta > 0$, we can get

$$\epsilon\left(\eta - \frac{L_1\eta^2}{2}\right) - \frac{L_1 E \eta^2 \sigma^2}{2} - \eta\delta^2 > 0. \tag{33}$$

Solving the above inequality yields

$$\eta < \frac{2\left(\epsilon - \delta^2\right)}{L_1\left(\epsilon + E\sigma^2\right)}. \tag{34}$$

Since $\epsilon, L_1, \sigma^2, \delta^2$ are all constants greater than 0, $\eta$ has solutions. Therefore, when the learning rate $\eta$ satisfies the above condition, any client's local mixed complete heterogeneous model can converge. Notice that the learning rate of the local complete heterogeneous model involves $\eta_\omega, \eta_\theta, \eta_\varphi$, so it's crucial to set them reasonably to ensure model convergence. Since all terms on the right side of Eq. (31) except for $1/T$ are constants, hence FedARC's non-convex convergence rate is $\epsilon \sim \mathcal{O}(1/T)$.

### A.2. Additional Experimental Details

**Experimental Environment.** To comprehensively assess the performance of FedARC, we conduct extensive experiments comparing it with 9 HtFL baselines across 4 standard benchmarks. All experiments are implemented in PyTorch and executed on a computing platform equipped with 8 NVIDIA GeForce RTX 3090 GPUs with 48GB of memory.

**Models.** As shown in Table 6, we allocate the diverse set of neural network architectures to clients. The output layer (i.e., the final fully connected layer) is configured with different dimensions based on the target dataset, such as 10 for Cifar10 and 100 for Cifar100.

---

**Algorithm 1** FedARC

---

**Input:** $N$, total number of clients; $K$, number of selected clients in one round; $T$, total number of rounds; $\eta_\omega$, learning rate of heterogeneous local models; $\eta_\theta$, learning rate of local extractors; $\eta_\varphi$, learning rate of representation projector.

**Output:** Randomly initialize the global homogeneous small model $\mathcal{G}(\theta^0)$, client local heterogeneous models $[\mathcal{F}_0(\omega_0^0), ..., \mathcal{F}_{N-1}(\omega_{N-1}^0)]$ and local heterogeneous representation projectors $[\mathcal{P}_0(\varphi_0^0), ..., \mathcal{P}_{N-1}(\varphi_{N-1}^0)]$.

1: **for** each round $t = 1, 2, ..., T-1$ **do**
2:     // **Server Side:**
3:     $\mathcal{S}^t \leftarrow$ Randomly sample $K$ clients from $N$ clients;
4:     Broadcast the global homogeneous feature extractor $\theta^{ex,t-1}$ to sampled $K$ clients;
5:     $\theta_k^{ex,t} \leftarrow$ **ClientUpdate**$(\theta^{ex,t-1})$
6:     /$*$ Aggregate Local Extractor $*$/
7:     $\theta^{ex,t} = \sum_{k=0}^{K-1} \frac{n_k}{n} \theta_k^{ex,t}$
8:     // **ClientUpdate:**
9:     Receive the global homogeneous feature extractor $\theta^{ex,t-1}$ from the server;
10:     **for** $k \in \mathcal{S}^t$ **do**
11:       /$*$ Local Iterative Training $*$/
12:       **for** $(\boldsymbol{x}_i, y_i) \in \mathcal{D}_k$ **do**
13:         $\mathcal{Z}_i^{\mathcal{F}_k} = \mathcal{F}_k^{ex}(\boldsymbol{x}_i; \omega_k^{ex,t-1}), \mathcal{Z}_i^{\mathcal{G}} = \mathcal{G}^{ex}(\boldsymbol{x}_i; \theta_k^{ex,t-1})$;
14:         $\mathcal{Z}_i = \mathcal{P}_k\left(\mathcal{Z}_i^{\mathcal{F}_k} \circ \mathcal{Z}_i^{\mathcal{G}}; \varphi_k^{t-1}\right)$;
15:         $\widetilde{\mathcal{Z}}_i^{\mathcal{F}_k} = \mathcal{Z}_i + \bar{\mathcal{R}}_i^{\mathcal{F}_k}, \quad \widetilde{\mathcal{Z}}_k^{\mathcal{G}} = \mathcal{Z}_i^{1:d_2} + \bar{\mathcal{R}}_i^{\mathcal{G}}$;
16:         $\hat{y}_i^{\mathcal{F}_k} = \mathcal{F}_k^{hd}(\widetilde{\mathcal{Z}}_i^{\mathcal{F}_k}; \omega^{hd,t-1}), \quad \hat{y}_i^{\mathcal{G}} = \mathcal{G}^{hd}(\widetilde{\mathcal{Z}}_i^{\mathcal{G}}; \theta^{hd,t-1})$;
17:         $\ell_i^{\mathcal{F}_k} = \ell_{ce}(\hat{y}_i^{\mathcal{F}_k}, y_i) + \lambda \cdot \ell_{mse}(\bar{\mathcal{Z}}_i^{\mathcal{F}_k}, \bar{\mathcal{Z}}^{g,1:d_1}), \quad \ell_i^{\mathcal{G}} = \ell_{ce}(\hat{y}_i^{\mathcal{G}}, y_i) + \lambda \cdot \ell_{mse}(\bar{\mathcal{Z}}_i^{\mathcal{G}}, \bar{\mathcal{Z}}^{g,1:d_2})$;
18:         $\ell_i = \alpha_i^{\mathcal{F}_k} \cdot \ell_i^{\mathcal{F}_k} + \alpha_i^{\mathcal{G}} \cdot \ell_i^{\mathcal{G}}$.
19:         $\omega_k^t \leftarrow \omega_k^{t-1} - \eta_\omega \nabla \ell_i, \quad \theta_k^t \leftarrow \theta^{t-1} - \eta_\theta \nabla \ell_i, \quad \varphi_k^t \leftarrow \varphi_k^{t-1} - \eta_\varphi \nabla \ell_i$;
20:       **end for**
21:       Upload updated local extractor $\theta_k^{ex,t}$ to the server.
22:     **end for**
23: **end for**
24: **return** heterogeneous local complete models $[\mathcal{F}_0(\omega_0^{T-1}, ..., \mathcal{F}_{N-1}(\omega_{N-1}^{T-1})]$.

---

**Datasets.** We provide 6 datasets across three modalities and three data heterogeneity scenarios. Specifically, we list all 6 datasets as follows:

1. **Cifar10**: Modality: image, Scenario: label skew, Description: 60K common images across 10 classes.

2. **Cifar100**: Modality: image, Scenario: label skew, Description: 60K common images across 100 classes.

3. **Flowers102**: Modality: image, Scenario: label skew, Description: 8K flower images across 102 classes.

4. **Tiny-ImageNet**: Modality: image, Scenario: label skew, Description: 100K common images across 200 classes.

5. **DomainNet**: Modality: image, Scenario: feature shift, Description: 600K images across 6 domains and 345 classes.

6. **AG News**: Modality: text, Scenario: label skew, Description: 127K articles across 4 classes.

**Heterogeneous Model Architectures** We adopt a widely recognized strategy for selecting model architectures, prioritizing official implementations, architectural diversity, and varying representational capacities. Through an extensive survey, we have incorporated 40 heterogeneous model architectures into HtFLlib, systematically organized into 19 distinct groups. Each group is designated for a specific experiment, with $X$ indicating the level of model heterogeneity—higher values of $X$ correspond to greater diversity in HtFE/HtM/HtC$_X^{dom}$. The full composition of all 12 model groups is detailed below:

1. **HtFE$^{img}_2$**: 4-layer CNN, and ResNet18.

2. **HtFE$^{img}_3$**: ResNet10, ResNet18, and ResNet34.

3. **HtFE$^{img}_5$**: GoogLeNet, MobileNetV2, ResNet18, ResNet34, and ResNet50.

4. **HtFE$^{img}_9$**: ResNet4, ResNet6, ResNet8, ResNet10, ResNet18, ResNet34, ResNet50, ResNet101, and ResNet152.

*Table 6.* The model architectures in HtFE$^{img}_8$ model group. "[5 × 5, 32]" represents a convolutional layer with kernel size 5×5 and output channel 32; "2 × 2 max pool" is a max pooling layer with kernel size 2×2. "B" is short for billion.

| Model | Sequentially Connected Feature Extractors | FLOPs |
|---|---|---|
| 4-layer CNN | [5 × 5, 32], 2 × 2 max pool, [5 × 5, 64], 2 × 2 max pool, 512-d fc | 0.013B |
| GoogLeNet | [7 × 7, 64], 3 × 3 max pool, Inception ×9, Inception ×5, Inception ×2, 1024-d fc | 1.530B |
| MobileNetV2 | [3 × 3, 32], 3 × 3 max pool, Bottleneck ×17, Bottleneck ×32, Bottleneck ×7, Bottleneck ×1, 1280-d fc | 0.314B |
| ResNet18 | [7 × 7, 64], 3 × 3 max pool, BasicBlock ×2, BasicBlock ×2, BasicBlock ×2, BasicBlock ×2, 512-d fc | 0.117B |
| ResNet34 | [7 × 7, 64], 3 × 3 max pool, BasicBlock ×3, BasicBlock ×4, BasicBlock ×6, BasicBlock ×3, 512-d fc | 0.218B |
| ResNet50 | [7 × 7, 64], 3 × 3 max pool, Bottleneck ×3, Bottleneck ×4, Bottleneck ×6, Bottleneck ×3, 2048-d fc | 1.305B |
| ResNet101 | [7 × 7, 64], 3 × 3 max pool, Bottleneck ×3, Bottleneck ×4, Bottleneck ×23, Bottleneck ×3, 2048-d fc | 2.532B |
| ResNet152 | [7 × 7, 64], 3 × 3 max pool, Bottleneck ×3, Bottleneck ×8, Bottleneck ×36, Bottleneck ×3, 2048-d fc | 5.330B |

5. **Res34-HtC$^{img}_4$**: ResNet34 with 4 types of headers.

6. **HtFE$^{img}_8$-HtC$^{img}_4$**: HtFE$^{img}_8$ with 4 types of headers.

7. **HtM$^{img}_{10}$**: HtFE$^{img}_8$ plus ViT-B/16 and ViT-B/32.

### A.3. Additional Experimental Results

**Robustness to Non-IIDness by Class Allocation.** To assess the effect of class-level heterogeneity, we systematically vary the number of classes assigned to each client under cross-device setting ($N = 50$, $\rho = 20\%$): Cifar10 uses 2/10 classes per client and Cifar100 uses 10/100. Fewer classes per client induce stronger non-IIDness. As shown in the upper panels of Fig. 6, FedARC outperforms best-performing baseline (FD) across all class-imbalance levels, maintaining a clear advantage as label space per client narrows.

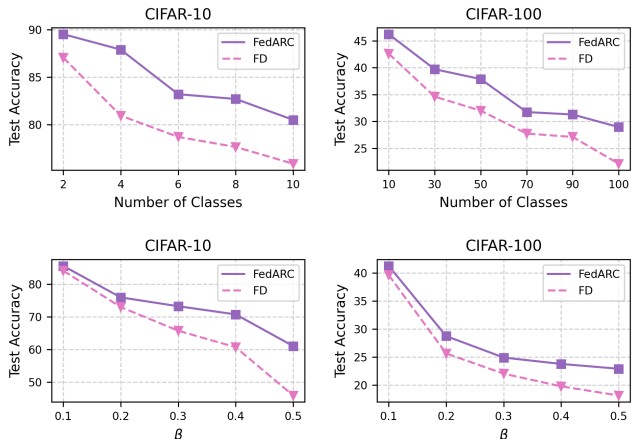

*Figure 6.* Robustness to non-IIDness (Class & Dirichlet).

**Robustness to Participation Rates.** We test the robustness of FedARC and FD against different client participant rates $\rho \in \{0.1, 0.3, 0.5, 0.7, 0.9, 1.0\}$ with $N = 100$ on Cifar10/100 (non-IID: 2/10 and 10/100). Across all $\rho$, FedARC remains superior to FD, with larger margins on the more challenging Cifar100 (Fig. 6). Notably, as $\rho$ increases, more clients per round provide more IID local data, steering updates toward a globally averaged model—improving generalization while diluting personalization.

*Table 7.* Test accuracy (%) on AG News in the cross-silo setting with various model heterogeneity.

|  | HtFE$^{txt}_2$ | HtFE$^{txt}_4$ | HtFE$^{txt}_6$ |
|---|---|---|---|
| FD | 91.63±0.12 | 79.06±0.21 | 87.89±0.15 |
| FedProto | 51.76±0.05 | 34.21±0.17 | 49.07±0.14 |
| FedTGP | 48.23±0.12 | 64.29±0.19 | 66.82±0.17 |
| LG-FedAvg | 84.43±0.08 | 75.46±0.23 | 72.19±0.18 |
| sFedGen | 84.26±0.05 | 82.03±0.26 | 78.08±0.23 |
| FedGH | 86.05±0.03 | 77.75±0.21 | 79.81±0.17 |
| pFedES | 86.69±0.12 | 88.92±0.21 | 89.01±0.18 |
| FedKD | 89.23±0.02 | 89.16±0.26 | 88.71±0.04 |
| FedMRL | 85.92±0.10 | 88.51±0.17 | 89.29±0.15 |
| FedARC | **93.52±0.09** | **91.17±0.16** | **91.96±0.11** |

**Impact of Model Heterogeneity.** From Table 7, we note three trends. (1) For methods that share only part of the parameters, accuracy typically declines as architectural heterogeneity increases. (2) By contrast, mutual-distillation and prototype-sharing approaches do not show a strictly monotonic drop with greater heterogeneity. (3) Beyond heterogeneity itself, the effectiveness of feature extraction strongly affects prototype-based methods: on text, FedProto and FedTGP lag behind their image-task counterparts. This indicates that, in NLP, architectural differences lead to divergent processing and context-modeling behaviors, making a unified feature space harder to align across clients. Consequently, aligning in the logit space is generally more reliable and efficient than direct feature-space alignment for text.

*Table 8.* Test accuracy (%) on Cifar100 in the cross-silo setting using HtFE$^{img}_8$ with different values of $\beta$. The results in "()" indicate the total number of converged rounds. We omit error bars here due to limited space.

|  | $\beta = 0.01$ | $\beta = 0.05$ | $\beta = 0.1$ | $\beta = 0.5$ | $\beta = 1$ |
|---|---|---|---|---|---|
| FD | 68.10(322) | 56.74(181) | 42.67(208) | 23.31(153) | 17.25(269) |
| FedProto | 60.26(562) | 46.69(411) | 36.12(556) | 19.41(581) | 12.81(391) |
| FedTGP | 67.82(226) | 55.81(308) | 40.69(211) | 22.87(217) | 18.30(263) |
| LG-FedAvg | 66.46(172) | 54.22(192) | 40.16(184) | 21.93(265) | 15.81(138) |
| FedGen | 66.16(161) | 52.10(165) | 38.68(158) | 21.53(141) | 15.45(158) |
| FedGH | 66.32(158) | 53.97(233) | 41.61(219) | 21.82(228) | 15.57(196) |
| pFedES | 68.53(288) | 55.83(210) | 42.37(201) | 23.28(175) | 17.83(322) |
| FedKD | 65.93(279) | 56.21(189) | 41.91(191) | 22.35(160) | 18.21(279) |
| FedMRL | 68.92(188) | 55.46(173) | 42.57(171) | 23.41(145) | 18.11(421) |
| FedARC | **72.52(140)** | **59.49(155)** | **44.21(147)** | **25.61(132)** | **20.17(164)** |

**Impact of Data Heterogeneity.** Table 8 reports accuracy and convergence rounds (in parentheses) on Cifar100 with HtFE$^{img}_8$ across different $\beta$. Prototype-sharing methods degrade markedly and often require many more rounds under stronger heterogeneity (e.g., FedProto), consistent with the fact that aggregating class means assumes a roughly aligned feature space—under label/feature skew the global prototypes become biased and less representative of local decision boundaries. By contrast, methods with server-side regularization or synthetic signals (e.g., FedTGP's margin-aware anchors; FedGen's generator-based alignment) tend to maintain more stable convergence as they partially decouple local updates from idiosyncratic client data. More broadly, theory shows that non-IID partitions induce gradient dissimilarity and client drift, slowing or destabilizing FedAvg-style training and increasing the communication rounds needed to reach a target accuracy; normalization/variance-reduction remedies were proposed exactly to counter this effect. Within this landscape, FedARC retains clear accuracy margins across all $\beta$ while converging among the fastest: its projection-based fusion plus alignment explicitly reduces cross-client representation mismatch, which mitigates drift and lowers the rounds needed to reach high accuracy.

*Table 9.* Test accuracy (%) on Cifar100 using HtFE$^{img}_8$ with 50/100/200 clients under partial participation ($\rho$).

| | $\rho = 50\%$ | | | $\rho = 10\%$ | |
| --- | --- | --- | --- | --- | --- |
| | 50 Clients | 100 Clients | 200 Clients | 100 Clients | 500 Clients |
| FD | 38.48±0.31 | 35.65±0.23 | 32.07±0.09 | 40.41±0.51 | 25.56±0.18 |
| FedProto | 20.20±0.48 | 18.81±0.51 | 15.25±0.42 | 20.76±0.92 | 12.07±0.23 |
| FedTGP | 36.71±0.21 | 35.87±0.28 | 27.81±0.63 | 29.53±0.51 | 24.51±0.31 |
| LG-FedAvg | 36.42±0.11 | 35.08±0.42 | 27.78±0.08 | 37.87±0.26 | 24.82±0.21 |
| FedGen | 35.02±0.23 | 33.54±0.23 | 26.91±0.21 | 33.34±0.48 | 22.05±0.33 |
| FedGH | 36.89±0.08 | 35.21±0.14 | 30.14±0.35 | 37.43±0.76 | 23.32±0.21 |
| pFedES | 37.92±0.29 | 35.62±0.62 | 34.32±0.52 | 39.49±0.32 | 25.17±0.28 |
| FedKD | 37.62±0.21 | 35.08±0.56 | 33.23±0.48 | 35.68±0.25 | 25.66±0.10 |
| FedMRL | 37.89±0.21 | 36.77±0.55 | 31.99±0.72 | 39.52±0.29 | 25.79±0.12 |
| FedARC | **40.56±0.10** | **38.12±0.35** | **36.56±0.21** | **41.74±0.18** | **28.20±0.14** |

**Impact of Client Participation.** We benchmark all methods with 50, 100, and 200 clients under partial participation ($\rho < 100\%$). From Table 9: (1) Accuracy drops for every baseline as the client pool grows, because partitioning Cifar100 across more clients leaves fewer samples per client, weakening local updates and, in turn, aggregation. (2) FedMRL benefits from jointly training a small global model with local models and therefore degrades less at low $\rho$, alleviating under-aggregation. (3) FedTGP remains competitive at 100 clients with $\rho = 50\%$, but deteriorates markedly at $\rho = 10\%$, where per-round participation is too limited to maintain reliable prototype guidance. Crucially, FedARC not only preserves its edge but widens it as the scenario becomes more dynamic. Despite larger participation size and unstable participation, FedARC remains the top performer. This indicates that our first-order semantic anchors computed once at initialization, smoothed with a light EMA, and coupled with residual compensation and $\lambda$ warm-up—do not lag or drift in highly dynamic environments.

## A.4. Knowledge Fusion Mechanisms under Feature Shift

As mentioned in the main paper, intra-client and inter-client knowledge fusion foster model-level and client-level collaboration, enabling the use of both domain-invariant (global) and domain-specific (local) information for better adaptation under data and model heterogeneity. We conduct in-depth ablations on the challenging DomainNet benchmark to probe these mechanisms at a fine-grained level. Here, clients have an identical number of labels but differ in the features of their data (e.g., sketch images and painting images), a canonical stress test for representation shift.

*Table 10.* Ablation Study of Intra-client Knowledge Transfer on DomainNet using HtFE$^{img}_5$. StopGrad "X"→Fusion detaches gradients from "X" before the projector while keeping its forward features unchanged.

| Method (Intra) | Clipart | Infograph | Painting | Quickdraw | Real | Sketch | Avg |
| --- | --- | --- | --- | --- | --- | --- | --- |
| w/o Intra | 25.16±0.83 | 20.11±1.96 | 22.17±1.02 | 54.28±0.80 | 28.15±1.04 | 25.62±0.85 | 29.25±0.69 |
| StopGrad L→Fusion | 28.41±1.05 | 19.82±1.22 | 24.59±1.19 | 57.02±0.46 | 31.05±0.99 | 28.71±0.98 | 31.60±0.33 |
| StopGrad G→Fusion | 30.52±0.65 | 20.26±0.67 | 24.71±1.11 | 58.46±0.51 | **33.54±0.41** | 28.83±1.82 | 32.72±0.23 |
| w/o Residual | 29.97±0.42 | **20.89±0.80** | 23.72±0.76 | 57.98±0.33 | 32.42±1.12 | 27.84±0.56 | 32.14±0.59 |
| **FedARC** | **30.76±0.61** | 20.67±1.13 | **26.19±0.69** | **60.08±0.58** | 32.74±0.81 | **29.32±0.76** | **33.30±0.50** |

**Ablation Study in Intra-client Knowledge Fusion.** To isolate intra-client fusion, we disable all inter-client operations and compare five variants (Table 10): (i) w/o Intra, which keeps the forward fusion but detaches both branches before the projector—no gradient flows back into either extractor; (ii) StopGrad L→Fusion, which detaches the local branch only—forward features are unchanged but fusion-path gradients do not update the local extractor (blocking global→local updates); (iii) StopGrad G→Fusion, which detaches the global branch only—forward features are unchanged but fusion-path gradients do not update the global extractor (blocking local→global updates); (iv) w/o Residual Compensation, which removes the client-level additive offsets applied to the fused features before the two headers; (v) FedARC, the default where features from the global homogeneous and local heterogeneous extractors are concatenated and fused by the projector, and gradients flow back into both branches. Throughout, architecture, losses, and hyper-parameters are unchanged; we alter only the direction of gradient flow (and whether residual offsets are present) to probe training-time transfer rather than ensembling. Using stop-gradient to control directional signal while keeping forward paths intact follows standard Siamese/self-supervised practice (e.g., SimSiam/BYOL).

(1) w/o Intra yields the lowest accuracy on most domains, confirming that interaction between the local and global extractors through the projector is essential for learning useful fused representations. (2) With StopGrad L→Fusion (blocking global→local updates), we observe larger drops on domains where personalization carries more weight (e.g., Quickdraw/Sketch), indicating that the local branch benefits materially from domain-invariant cues provided by the global branch. (3) With StopGrad G→Fusion (blocking local→global updates), a distinctive pattern emerges: relative to w/o Intra, several other domains can improve because the remaining trainable local extractor specializes more strongly to its home domain once the global branch is insulated; however, cross-domain generalization deteriorates, so the overall average remains below FedARC. (4) w/o Residual further shows that client-level additive offsets contribute additional gains—particularly on strongly shifted domains (e.g., Quickdraw/Sketch)—consistent with residuals acting as lightweight feature-space calibration. These directional effects align with complementary roles: the local branch contributes domain-specific signals, the global branch supplies domain-invariant structure, and bidirectional fusion plus residual calibration delivers the best aggregate performance.

*Table 11.* Ablation Study of Inter-client Knowledge Transfer on DomainNet using HtFE$^{img}_5$.

| Method (Inter) | Clipart | Infograph | Painting | Quickdraw | Real | Sketch | Avg |
|---|---|---|---|---|---|---|---|
| w/o Inter | 25.25±1.42 | 18.01±2.04 | 22.83±0.96 | 53.98±0.83 | 27.37±1.12 | 26.10±0.81 | 28.92±0.65 |
| No Aggregation only | 29.98±0.96 | 18.63±1.03 | 23.10±1.34 | 56.58±0.67 | 28.52±0.89 | 26.97±1.54 | 30.63±0.53 |
| No Anchor only | 30.36±0.83 | **18.99±0.89** | **24.46±1.28** | 58.11±0.52 | 31.31±0.56 | 28.58±1.64 | 31.97±0.68 |
| **FedARC** | **30.93±0.73** | 18.78±0.73 | 24.17±0.89 | **61.82±0.28** | **33.54±0.72** | **29.86±0.67** | **33.16±0.24** |

**Ablation Study in Inter-client Knowledge Fusion.** Here we disable intra-client fusion and examine inter-client exchange (Table 11): (i) w/o Inter (no aggregation of the global extractor; no semantic-anchor broadcast); (ii) No Aggregation only (semantic anchor on; aggregation off); (iii) No Anchor only (aggregation on; anchor off); and (iv) FedARC (Inter=ON) (both on).

Removing inter-client exchange (w/o Inter) degrades accuracy across all domains, underscoring the need for cross-client knowledge in the feature-shift regime of DomainNet. Re-enabling aggregation or anchor alignment alone yields clear gains; the full configuration (aggregation + anchor) performs best on nearly every domain, showing that (a) the server-side aggregation injects transferable global semantics and (b) the semantic anchor further reduces inter-client representational drift so that heterogeneous clients can exploit the shared subspace more effectively. These trends are consistent with prior analyses that separate model-level (within-client) and client-level (cross-client) cooperation when feature shift is present.

*Table 12.* Test accuracy (%) under pathological settings and practical scenario using HtFE$^{img}_8$

| Settings | | Pathological Setting | | | Practical Scenario | |
|---|---|---|---|---|---|---|
| | Cifar10 | Cifar100 | Flowers102 | Tiny* | $\rho = 0.5$ | $\rho = 1.0$ |
| FD | 88.76±0.06 | 56.47±0.24 | 59.75±0.32 | 32.83±0.35 | 37.17±0.27 | 38.81±0.21 |
| FedProto | 85.37±0.17 | 51.59±0.27 | 54.27±0.19 | 28.29±0.34 | 28.20±0.35 | 27.86±0.27 |
| FedTGP | 88.05±0.24 | 56.35±0.31 | 62.48±0.29 | 33.65±0.29 | 36.89±0.43 | 38.71±0.35 |
| LG-FedAvg | 87.04±0.28 | 57.23±0.62 | 59.95±0.27 | 33.40±0.19 | 34.23±0.31 | 33.23±0.27 |
| FedGen | 86.82±0.34 | 55.32±0.32 | 60.39±0.14 | 30.08±1.07 | 32.53±0.27 | 34.89±0.12 |
| FedGH | 87.57±0.14 | 57.39±0.18 | 60.24±0.32 | 33.84±0.36 | 35.39±0.23 | 33.27±0.11 |
| pFedES | 88.76±0.21 | 59.81±0.27 | 61.04±0.28 | 33.79±0.22 | 38.45±0.24 | 37.36±0.29 |
| FedKD | 88.32±0.32 | 56.75±0.25 | 56.82±0.37 | 34.46±0.34 | 35.51±0.21 | 34.27±0.34 |
| FedMRL | 88.89±0.21 | 59.92±0.30 | 61.72±0.48 | 34.02±0.32 | 38.97±0.26 | 37.54±0.14 |
| FedARC | **90.27±0.11** | **62.01±0.12** | **67.59±0.27** | **35.47±0.21** | **40.81±0.14** | **39.07±0.12** |

## A.5. More Experimental Results On Benchmark Datasets

**Pathological setting.** For the pathological setting, following FedAvg, we distribute non-redundant and unbalanced data of 2/10/10/20 classes to each client from a total of 10/100/102/200 classes on Cifar10/Cifar100/Flowers102/Tiny-Ima geNet datasets, respectively. This stresses class-skewed learning without domain confounders. Across all four datasets, FedARC is consistently best: it surpasses the strongest baselines that operate at the head/prototype/logit level (e.g., FedTGP/FedMRL) and those that couple global–local models via KD or shared headers. We attribute the gains to operating in feature space: concatenation→projection to a fused space, slice-specific residual corrections ($1 : d_1$ and $1 : d_2$), and semantic-anchor alignment that stabilizes the shared prefix subspace before either head consumes the features.

**Practical scenario (new clients / cold-start).** To simulate a practical scenario with new clients joining for future HtFL, we perform method-specific local training for 10 epochs on new participants for warming up after their local models are initialized by the learned global model. Specifically, using Cifar100 and HtFE$^{img}_8$, we conduct FL on 80 old clients ($\rho = 0.5$ or $\rho = 0.1$) and evaluate accuracy on 20 new joining clients after warming up. FedARC again leads at both $\rho$ values, indicating stronger fast alignment for new participants. Mechanistically, the global semantic anchor provides a client-invariant first-order target in the shared subspace, while client-specific residuals quickly absorb local drifts—yielding better cold-start behavior than methods that rely solely on head/prototype/logit transfer. (Warm-up protocols for newly joined clients are standard in FL and help stabilize personalization before full participation.)

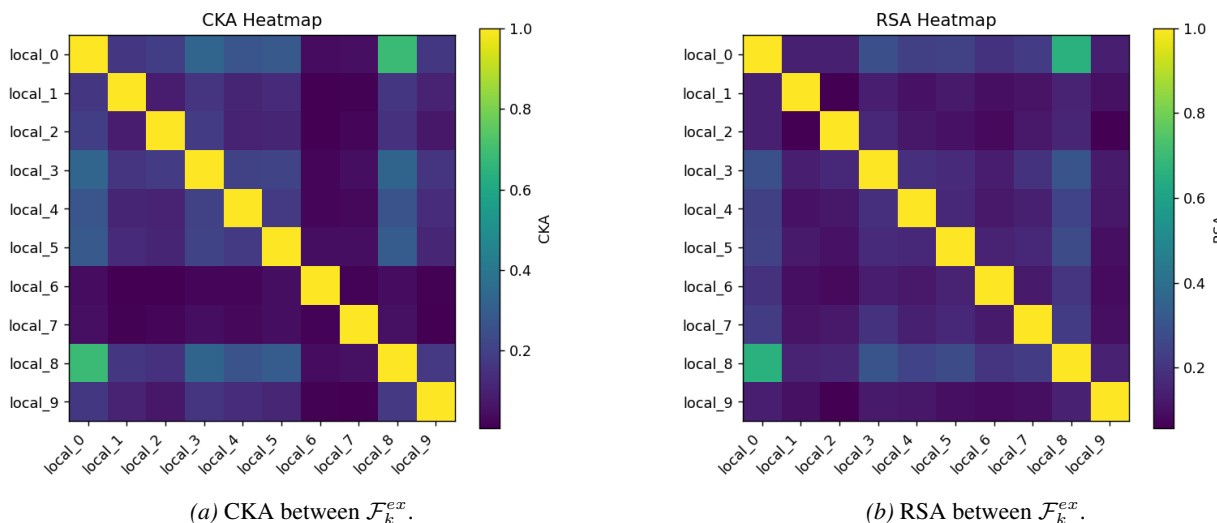

*(a)* CKA between $\mathcal{F}^{ex}_k$.  *(b)* RSA between $\mathcal{F}^{ex}_k$.

*Figure 7.* Cross-client representation similarity of the local heterogeneous feature extractors $\mathcal{F}^{ex}_k$.

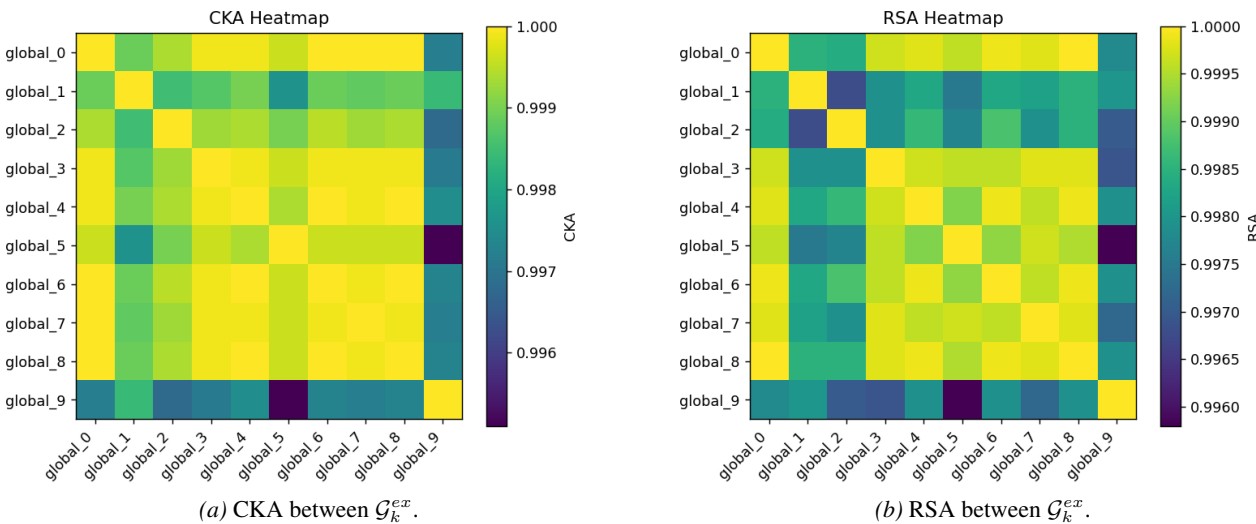

*(a)* CKA between $\mathcal{G}^{ex}_k$.  *(b)* RSA between $\mathcal{G}^{ex}_k$.

*Figure 8.* Cross-client representation similarity of the global homogeneous feature extractors $\mathcal{G}^{ex}_k$.

## A.6. Additional CKA/RSA analysis

Fig. 7 and Fig. 8 report cross-client CKA (left) and RSA (right) similarities of the local and global feature extractors, respectively, computed on a shared probe set. Each heatmap entry $(i, j)$ measures the similarity between the encoder of client $i$ and that of client $j$.

In the **local** encoder plots (Fig. 7), the diagonal entries are by construction 1.0, while most off-diagonal CKA values stay in a low range (approximately 0.05–0.30) and the corresponding RSA scores are similarly small. This indicates that different local branches $\mathcal{F}_k^{ex}$ learn clearly distinct representations, reflecting client-specific distribution shifts rather than collapsing to a single shared solution. Only a few client pairs (e.g., client 0 vs. 8) show moderately higher similarity, suggesting partial overlap but still far from full alignment.

In contrast, the **global** encoder plots (Fig. 8) exhibit uniformly high off-diagonal similarities: all cross-client CKA and RSA values are very close to 1.0. This means that the global extractors $\mathcal{G}_k^{ex}$ across clients implement almost identical representation geometries, as expected from a shared federated model that captures domain-invariant structure.

Taken together, these four heatmaps provide empirical evidence for our design assumption: the split encoders do not redundantly learn the same features. Instead, the global branch converges to a highly aligned, shared representation across clients, while the local branches remain diverse and client-specific, which in turn justifies the need for the proposed adaptive residual compensation to bridge these heterogeneous local spaces with the shared global representation.

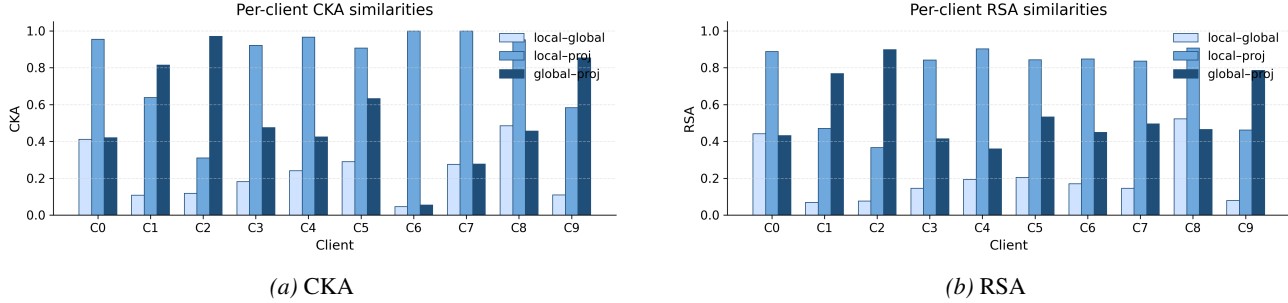

*(a)* CKA             *(b)* RSA

*Figure 9.* Per-client similarities among the local encoder, the global encoder, and the projector, measured by CKA (left) and RSA (right).

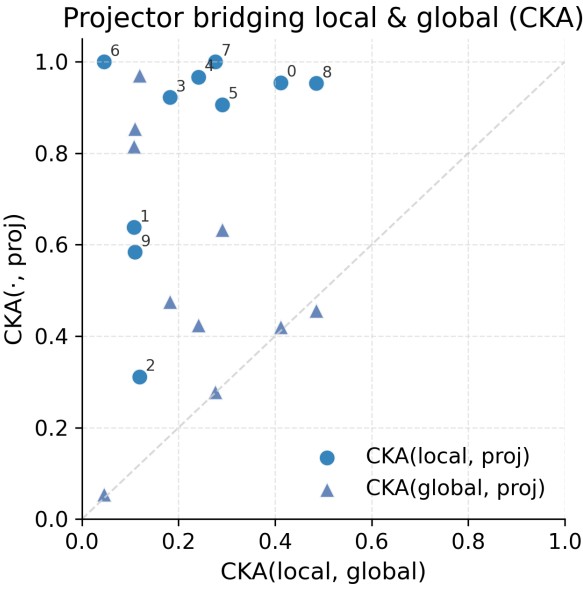

*Figure 10.* Projector similarity vs. local–global CKA for each client (projector bridges the two encoders).

### A.7. Additional CKA/RSA analysis of local, global, and projector branches

For each client $k$, we further compare the representations from the local encoder $\mathcal{F}_k^{ex}$, the shared global encoder $\mathcal{G}^{ex}$, and the projector $\mathcal{P}_k$ on that client's own data. Figures 9 report, the per-client CKA and RSA similarities between the three branches: local-global, local-proj, and global-proj. We observe that local-global similarities are consistently the lowest across clients (mean CKA $\approx 0.23$, RSA $\approx 0.21$), while pairs involving the projector are substantially higher (local–proj:

mean CKA $\approx 0.82$, RSA $\approx 0.74$; global–proj: mean CKA $\approx 0.54$, RSA $\approx 0.56$). This pattern confirms that the split encoders indeed learn distinct representations, with the local branch capturing client-specific signals that are not trivially aligned with the global branch, and the projector acting as a learned adapter that remains close to the local space while still maintaining non-trivial alignment with the global space.

Fig. 10 visualizes this effect more directly: each point corresponds to a client, with the $x$-axis showing CKA(local, global) and the $y$-axis showing either CKA(local, proj) or CKA(global, proj). Almost all points lie well above the $y = x$ diagonal, indicating that each encoder is much more similar to the projector than to the other encoder itself. In other words, the projector learns to bridge the two representation spaces rather than collapsing them into a single shared solution. Together with the directional stop-gradient ablations in the main text, these results provide empirical support for our conceptual view: the local and global branches specialize to client-specific vs. shared factors, while the adaptive residual compensation (implemented via $\mathcal{P}_k$ and residuals) aligns them into a unified representation used for prediction.

*Table 13.* Test accuracy (%) on DomainNet with HtFE$^{img}_8$.

| Method | Clipart | Infograph | Painting | Quickdraw | Real | Sketch | Avg |
|---|---|---|---|---|---|---|---|
| FD | 26.05±0.69 | 17.01±1.12 | 24.16±0.72 | 59.86±0.42 | 31.70±0.36 | 21.40±0.43 | 30.03±0.62 |
| FedProto | 19.22±0.53 | 13.79±1.06 | 19.76±0.47 | 55.11±0.50 | 22.25±0.53 | 16.80±0.38 | 24.49±0.58 |
| FedTGP | 21.82±0.45 | 15.88±0.99 | 20.24±0.51 | 59.20±0.54 | 23.70±0.43 | 18.40±0.26 | 26.54±0.53 |
| LG-FedAvg | 27.69±0.51 | 16.54±0.56 | 24.07±0.34 | 56.03±0.39 | 28.12±0.41 | 21.05±0.43 | 28.91±0.44 |
| FedGen | 27.62±0.31 | 15.49±0.42 | 23.82±0.27 | 56.09±0.29 | 28.92±0.30 | 19.76±0.29 | 28.62±0.31 |
| FedGH | 25.52±0.22 | 16.15±0.34 | 22.77±0.17 | 56.66±0.23 | 27.50±0.22 | 19.20±0.20 | 27.97±0.23 |
| pFedES | 25.91±0.70 | 17.07±0.91 | 24.20±0.41 | 59.90±0.42 | 29.74±0.44 | 21.44±0.40 | 29.71±0.55 |
| FedKD | 25.42±0.48 | 17.61±0.42 | 23.59±0.29 | 58.64±0.25 | 29.41±0.27 | 21.56±0.26 | 29.37±0.33 |
| FedMRL | 25.36±0.35 | 16.95±0.63 | 23.32±0.16 | 58.24±0.17 | 29.65±0.26 | 21.10±0.21 | 29.10±0.30 |
| FedARC | **30.54±0.69** | **20.13±1.14** | **25.50±0.36** | **61.62±0.41** | **32.63±0.79** | **29.28±0.70** | **33.28±0.68** |

## A.8. Extended Experiment on DomainNet

Table 13 reports the results on DomainNet using HtFE$^{img}_8$. FedARC achieves the best performance across all six domains and obtains the highest average accuracy. The improvement is particularly significant on Sketch and Clipart, indicating that FedARC better handles severe domain discrepancy and representation heterogeneity. Prototype- and distillation-based methods, such as FedProto and FedTGP, show relatively limited performance under strong heterogeneity, suggesting that global semantic alignment alone is insufficient in this setting. Other methods achieve more competitive results, but still struggle to consistently address cross-domain feature drift. The results demonstrate the effectiveness of adaptive residual compensation and semantic anchor alignment for HtFL.

*Table 14.* Average test accuracy (%) on four datasets in cross-silo and cross-device settings under label skew using HtFE$^{img}_8$.

| Settings | Cross-silo (N=10, $\rho$=100%, $\beta$=0.1) | | | | Cross-device (N=50, $\rho$=20%, $\beta$=0.1) | | | |
|---|---|---|---|---|---|---|---|---|
| Datasets | Cifar10 | Cifar100 | Flower102 | Tiny* | Cifar10 | Cifar100 | Flower102 | Tiny* |
| FedBN | 84.37±0.16 | **44.67±0.04** | 51.33±0.21 | 27.29±0.26 | 83.16±0.20 | 40.07±0.10 | 45.33±0.28 | 23.89±0.21 |
| FedBABU | 86.42±0.12 | 44.09±0.07 | 51.81±0.19 | 26.67±0.15 | 83.53±0.18 | 39.91±0.11 | 44.33±0.22 | 23.72±0.18 |
| FedCO2 | 86.08±0.12 | 42.97±0.21 | 50.44±0.17 | 28.97±0.21 | 84.07±0.18 | 37.16±0.28 | 45.35±0.22 | 25.18±0.19 |
| FedRoD | 83.84±0.22 | 41.86±0.25 | 49.90±0.13 | 28.74±0.18 | 79.14±0.31 | 36.16±0.21 | 44.81±0.26 | 24.46±0.35 |
| FedDRM | 71.71±0.17 | 31.44±0.02 | 42.05±0.15 | 20.10±0.18 | 70.06±0.21 | 30.13±0.23 | 40.33±0.23 | 18.01±0.20 |
| FedARC | **89.22±0.02** | 44.21±0.05 | **57.67±0.11** | **29.91±0.14** | **86.14±0.14** | **42.25±0.07** | **50.18±0.15** | **27.10±0.16** |

## A.9. Extended Experiment

As reported in Table 14, under the same HtFE$^{img}_8$ heterogeneous setting with label skew, FedARC consistently outperforms both FedCO2 (Cai et al., 2023) and Fed-RoD (Chen & Chao, 2022) across all four datasets and in both cross-silo (10 clients, full participation) and cross-device (50 clients, partial participation) regimes, with gains of roughly 1–7 % over the best baseline in each configuration; while FedCO2 and Fed-RoD coordinate global–local behavior mainly via prediction-level cooperation under homogeneous architectures, FedARC calibrates fused heterogeneous representations through residual compensation and semantic anchoring, yielding stronger accuracy under the same HtFL protocol. We clarify that FedBN and FedBABU were originally designed for homogeneous FL / personalization settings rather than native HtFL. Therefore,

when adapting them to the HtFL benchmark, we preserved their original mechanisms as faithfully as possible. FedBN remains competitive on Cifar100, where local BN effectively captures client-specific statistics, but its advantage weakens under more challenging cross-device partial participation since it cannot address cross-client semantic drift. FedBABU also performs strongly because its frozen head / shared representation learning strategy benefits complex multi-class tasks; however, its computation cost is substantially higher than most HtFL methods, limiting practicality in resource-constrained edge scenarios. Although FedDRM is an HtFL method, its official design relies on stable mini-batch statistics and reliable client-density estimation. Under the unified HtFL benchmark with smaller local batches and stronger heterogeneity, its routing branch becomes harder to optimize, leading to reduced robustness and accuracy.

*Table 15.* Robustness of FedARC under varying feature dimension settings $(d_1, d_2)$.

|  | $d_1 = 128$ | $d_1 = 256$ | $d_1 = 512$ | $d_1 = 1024$ |
|---|---|---|---|---|
| $d_2 = 128$ | 40.09±0.06 | 43.84±0.09 | 43.27±0.08 | 42.21±0.08 |
| $d_2 = 256$ | - | 42.86±0.07 | **44.21±0.05** | 42.65±0.08 |
| $d_2 = 512$ | - | - | 43.19±0.08 | 41.75±0.06 |
| $d_2 = 1024$ | - | - | - | 40.78±0.07 |

## A.10. Robustness to feature dimension variation

We observe that FedARC achieves consistently stable performance across different combinations of $(d_1, d_2)$, with only minor fluctuations. In particular, no clear degradation is observed when increasing either local or shared feature dimensions, suggesting that the method is not sensitive to specific feature dimensional settings. This empirically supports the robustness of FedARC under varying feature allocation schemes.

## A.11. Hyperparameter Sensitivity and Tuning

We adopt a per-architecture tuning and cross-dataset reuse policy to avoid dataset-specific over-tuning. Specifically, we perform a grid search once per backbone family to select the anchor weight $\lambda$ and moving-average momentum $\kappa$, then fix these values across all datasets for that architecture. The search space is defined as $\lambda \in \{0, 0.01, 0.1, 1, \ldots, 500\}$ and $\kappa \in \{0, 0.1, \ldots, 1.0\}$. Specifically, grid search is performed in the following search space:

| $\lambda$ | 0, 0.01, 0.1, 1, 5, 10, 20, 50, 100, 200, 500 |
|---|---|
| $\kappa$ | 0, 0.1, 0.2, 0.3, 0.4, 0.5, 0.6, 0.7, 0.8, 0.9, 1.0 |

Here, $\lambda$ balances the semantic-anchor MSE with the supervised loss; we observe that larger $\lambda$ helps counter significant inter-client semantic drift. The momentum $\kappa$ controls the update of client-side running statistics, similar to BatchNorm/EMA: a smaller $\kappa$ provides heavier smoothing to stabilize noisy batch means, while a larger $\kappa$ enables quicker adaptation. Empirically, FedARC exhibits a broad stable region and does not require fragile per-dataset retuning. Selected values: In the image classification tasks, we set $\lambda = 1, \kappa = 0.1$ for most backbones (e.g., $\text{HtFE}^{img}_X$, $\text{Res34-HtC}^{img}_4$, $\text{HtFE}^{img}_8\text{-HtC}^{img}_4$ and $\text{HtM}^{img}_{10}$); in the text classification tasks, we set $\lambda = 0.1, \kappa = 1$ for $\text{HtFE}^{txt}_X$.

## A.12. Data Distribution Visualization

We illustrate the data distributions (including training and test data) in our experiments here.

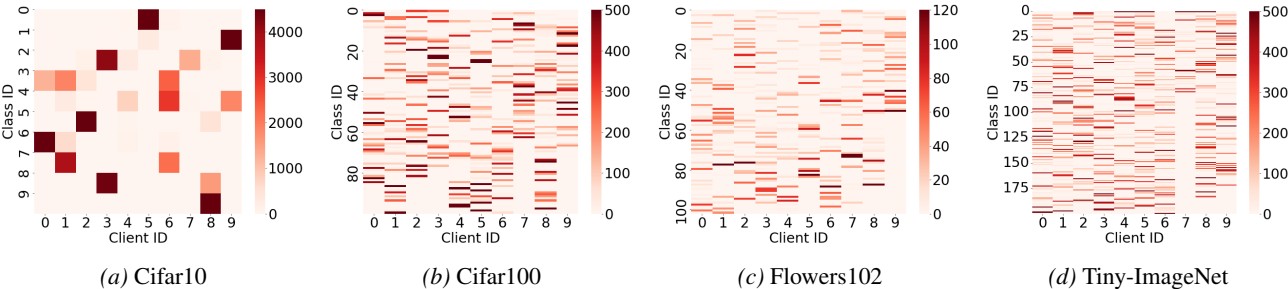

| *(a)* Cifar10 | *(b)* Cifar100 | *(c)* Flowers102 | *(d)* Tiny-ImageNet |

*Figure 11.* The data distributions of all clients on Cifar10, Cifar100, Flowers102, and Tiny-ImageNet, respectively, in the practical settings ($\beta = 0.1$). The depth of the color represents the number of samples.

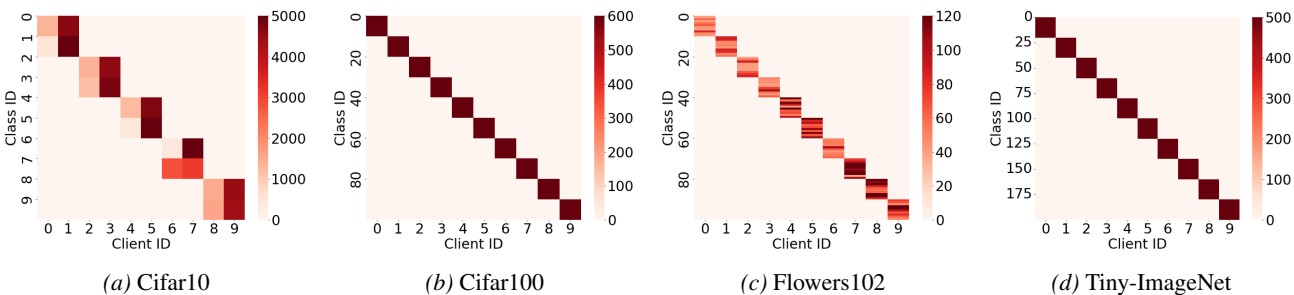

| *(a)* Cifar10 | *(b)* Cifar100 | *(c)* Flowers102 | *(d)* Tiny-ImageNet |

*Figure 12.* The data distributions of all clients on Cifar10, Cifar100, Flowers102, and Tiny-ImageNet, respectively, in the pathological settings. The depth of the color represents the number of samples.

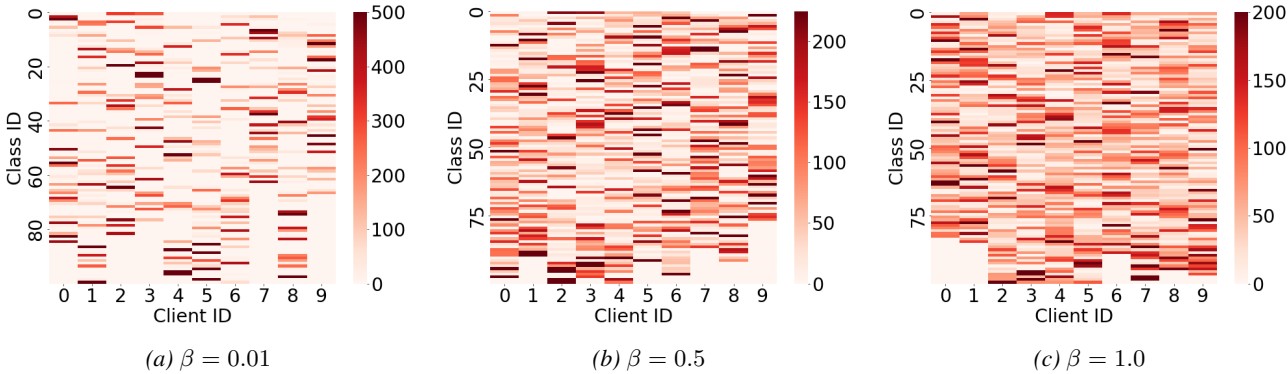

| *(a)* $\beta = 0.01$ | *(b)* $\beta = 0.5$ | *(c)* $\beta = 1.0$ |

*Figure 13.* The data distribution on all clients on Tiny-ImageNet in three additional practical settings. The depth of the color represents the number of samples. The degree of heterogeneity decreases as $\beta$ in $\text{Dir}(\beta)$ increases.

