# OpenReview forum: "FedARC: Anchor-Guided Residual Compensation for Data and Model Heterogeneous Federated Learning"
_ICML.cc/2026/Conference — ICML 2026 spotlight_

### Official Review · Reviewer_Zu6U · 2026-03-01

**Soundness:** 3
**Presentation:** 3
**Significance:** 3
**Originality:** 3
**Overall Recommendation:** 4
**Confidence:** 4

**Summary:**

This paper focuses on the heterogeneous federated learning (HtFL) under both data and model heterogeneity, especially in the cross-device settings where client participation is sparse. The authors introduce FedARC, a feature alignment framework that combines a shared lightweight homogeneous extractor and local heterogeneous extractors, along with adaptive residual compensation and semantic anchor alignment, to address the cross-client representation drift while preserving personalization. In this paper, the authors evaluate their method through extensive experiments on various datasets and model architectures, achieving strong performance compared with existing HtFL baselines.

**Compliance With Llm Reviewing Policy:**

Affirmed.

**Final Justification:**

After reading the rebuttal, I agree that the rebuttal helps clarify the method, including the adaptive anchor update strategy, more experimental analysis, and the privacy discussion. I decide to maintain my score.

**Key Questions For Authors:**

Some questions for authors:
1. It appears that the paper initializes the global semantic anchor only once and then fixes it. Could the semantic anchor be adaptively updated across rounds, and what implications would this have for stability?
2. How does FedARC scale under very large client populations (e.g., > 500 clients) with highly sparse participation?
3. Since these client-specific semantic statistics summarize local representations across training batches, could the authors clarify whether they may introduce additional privacy risks, such as leaking information about local data distributions or increasing the risk of representation reconstruction?
4. The paper provides a practical scenario with new clients joining after training. Would the authors mind explaining why FedARC appears to adapt well in this cold-start setting, and which component is most important for this adaptation?

**Limitations:**

yes

**Strengths And Weaknesses:**

Strengths:
1. The motivation and discussion of the limitations of previous HtFL methods are clear.
2. The problem setting is practically meaningful, and the paper directly addresses two key FL challenges caused by data and model heterogeneity, which are essential for real-world FL deployment.
The problem statement is practical and relevant, and the work tackles a real-world problem of HtFL due to data and model heterogeneity.
3. The proposed method FedARC is well-motivated and technically sound.
4. Extensive benchmarks and theoretical analysis are included to support the paper’s contributions.

Weaknesses:
1. The $\bar{Z}^g$ is fixed after initialization and does not appear to be updated during training. This might limit its ability to adapt to changes in the representation space. Additionally, the SAA (Semantic Anchor Alignment) based on MSE introduces additional computation. Exploring an adaptive anchor update strategy could further improve the method.
2. Further analysis on highly scaled FL scenarios (such as hundreds or thousands of clients) would help to further strengthen the practical relevance.
3. The paper makes use of client-specific semantic means to build global semantic anchors, but the privacy implications of these statistics are not discussed in much detail.
4. Interaction between clients and server is not clear. Section 2.1 and Figure 2 make it clear that the global homogeneous extractor $\mathcal{G}^{ex}$ is broadcast and aggregated, but it is not so clear how the semantic statistics used for constructing the global semantic anchor are communicated and synchronized across rounds.
5. Also a small typo appears in the caption of Figure 1, where "model/representationsmodule" seems to be missing a space. This likely should be "model/representations module".

---

> ### Author Rebuttal · Authors · 2026-03-31
>
> **W1/Q1**: Thank you for raising this important question. In FedARC, the global semantic anchor is initialized once and then kept fixed during training. This is a deliberate design choice to provide a stable target for cross-client alignment. Updating the anchor dynamically would introduce additional noise from heterogeneous clients, potentially leading to drifting targets and unstable optimization. This design is consistent with prior anchor-based approaches [1], where a stable reference is used to guide representation alignment, rather than continuously adapting to client-specific updates. If $\bar{Z}^g$ were updated online, the alignment target would drift together with local representations, making the projector $\mathcal{P}_k$ and residual corrections $(\mathcal{R}_i^{\mathcal{F}_k},\mathcal{R}_i^{\mathcal{G}})$ chase a moving target and weakening the consistency of the learned transformation. Moreover, our empirical results (Tables 10–11) show that the fixed-anchor design already achieves strong and stable performance. Our current design prioritizes stability and robustness, which is critical for consistent representation alignment.
>
> [1] FedSA: A Unified Representation Learning via Semantic Anchors for Prototype-based Federated Learning. AAAI 2025.
>
> **W2/Q2**: Thank you for this important suggestion. FedARC scales well to large and sparsely participating client populations. We further evaluate the setting with 500 clients and a low participation ratio $(\rho = 0.1)$, where FedARC consistently achieves the best performance:
>
> ||$\rho=0.1$|
> |-|-|
> ||500 Clients|
> |FD|25.56±0.18|
> |FedProto|12.07±0.23|
> |FedTGP|24.51±0.31|
> |LG-FedAvg|24.82±0.21|
> |FedGen|22.05±0.33|
> |FedGH|23.32±0.21|
> |pFedES|25.17±0.28|
> |FedKD|25.66±0.10|
> |FedMRL|25.79±0.12|
> |FedARC|**28.20±0.14**|
>
> FedARC follows the standard FL setting with client-independent local training and server-side aggregation, without introducing additional communication or synchronization overhead. The ARC and SAA modules operate locally in the representation space and require no extra client-to-client interaction, ensuring scalability as the number of clients increases. We will include these results and further clarify the scalability of FedARC in the revised manuscript.
>
> **W3/Q3**: Thank you for raising this important concern. In FedARC, the shared semantic information consists only of aggregated feature statistics rather than raw data, gradients, or model parameters. Specifically, each client maintains running feature means locally and contributes aggregated statistics to construct the global semantic anchor. These statistics are low-dimensional and non-invertible summaries, which limit the risk of reconstructing original data. Moreover, aggregation further reduces privacy exposure, as the global semantic anchor is obtained by averaging across multiple clients, making it difficult to infer any individual client's data distribution. Prior works have shown that privacy risks in FL mainly arise from gradient or model update sharing, which can enable reconstruction attacks [2], whereas aggregated statistics provide limited information for exploitation. Aggregated statistics (e.g., batch normalization statistics or prototypes) are therefore less sensitive than raw data or gradients. We will clarify this aspect in the revised manuscript.
>
> [2] FedInverse: Evaluating Privacy Leakage in Federated Learning. ICLR 2024.
>
> **W4**: Thank you for pointing this out. We clarify that semantic statistics are synchronized **only once at initialization**, rather than in every communication round. Specifically, the server initializes and broadcasts $\mathcal{G}^{ex}$; clients compute local semantic means in fused and shared prefix spaces; the server aggregates them to form the global anchor $(\bar{Z}^g,\bar{Z}^{g,1:d_2})$, which is broadcast and kept fixed during training. Subsequent training follows standard FL rounds, where only $\mathcal{G}^{ex}$ is communicated and aggregated, without further exchange of semantic statistics. This design is consistent with anchor-based FL approaches [1]. We will revise Section 2.1 to explicitly describe this protocol.
>
> **W5**: Thank you for your careful reading and for pointing out this typo. We will correct it in the revised manuscript.
>
> **Q4**: Thank you for your insightful question. FedARC performs effectively in cold-start settings as it learns a **shared representation space with a stable semantic reference**. The global semantic anchor provides a consistent alignment target, enabling new clients to align their representations without accessing previous data. Among all components, SAA is the most critical, as it enforces cross-client semantic consistency, while ARC further adjusts client-specific deviations. As shown in Tables 10–11, both alignment and correction are essential components, jointly enabling effective knowledge transfer within and across clients. We will further clarify this behavior in the revised manuscript.

---

> > ### Author Rebuttal · Reviewer_Zu6U · 2026-04-02
> >
> > After reading the rebuttal, I agree that the rebuttal helps clarify the method, including the adaptive anchor update strategy, more experimental analysis, and the privacy discussion. Overall, I appreciate the authors' clarifications, thanks for the rebuttal.

---

### Official Review · Reviewer_iRz5 · 2026-03-02

**Soundness:** 4
**Presentation:** 3
**Significance:** 3
**Originality:** 3
**Overall Recommendation:** 5
**Confidence:** 4

**Summary:**

The paper proposes FedARC, a heterogeneous federated learning (HtFL) framework to handle simultaneous data and model heterogeneity by improving cross-client representation alignment. FedARC fuses local and global features via a lightweight client-specific projector, applies adaptive residual compensation (ARC) to correct feature-level mismatches, and introduces semantic anchor alignment (SAA) to regularize client representations toward a global anchor in a shared subspace. Extensive experiments on multiple benchmarks including Cifar10, Cifar100, Tiny-ImageNet, DomainNet, Flowers102, and AGNews, demonstrate that FedARC outperforms existing methods, achieving accuracy improvements of up to 2.63% across different settings.

**Compliance With Llm Reviewing Policy:**

Affirmed.

**Final Justification:**

The authors have successfully addressed all my concerns, and I will maintain my positive score of 5.

**Key Questions For Authors:**

1. Could the authors elaborate on the robustness of FedARC to a wider range of feature dimensions? Specifically, how does the framework perform under extreme scaling variations between the local feature dimension $d_1$ and the global feature dimension $d_2$?

2. What systematic procedure was utilized to tune and select key hyperparameters (such as $\lambda$ and $\kappa$) across the diverse datasets and heterogeneous model architectures?

3. Could the authors provide the structural details for the lightweight client-specific projector $\mathcal{P}_k(\varphi_k)$ used in Eq. (5) for the $\mathbb{R}^{d_1+d_2} \rightarrow \mathbb{R}^{d_1}$ mapping?

**Limitations:**

yes

**Strengths And Weaknesses:**

Strengths:
1. The paper addresses a significant challenge in heterogeneous federated learning, effectively targeting the representation bias and representation degradation caused by the simultaneous presence of data and model heterogeneity.

2. The proposed FedARC framework is well motivated. The Adaptive Residual Compensation mechanism preserves client specific bias for adaptive feature space compensation. Concurrently, the Semantic Anchor Alignment module effectively regularizes representation means to encourage cross client consistency.

3. Extensive experimental validation strongly supports the core claims, demonstrating the method's strong empirical coverage and stability across multiple datasets, participation settings, and varying degrees of heterogeneity.

Weaknesses:

1. While the current experiments demonstrate exceptionally strong performance on various heterogeneous model settings, the robustness of FedARC to a wider range of feature dimensions, specifically variations between the local feature dimension $d_{1}$ and the global feature dimension $d_{2}$, is not yet fully characterized.

2. The paper does not fully specify the exact procedure for how hyperparameters, such as the regularization strength $\lambda$ and the momentum coefficient $\kappa$, are systematically selected across the different datasets and diverse heterogeneous models.

3. The framework heavily relies on a lightweight client-specific projector $\mathcal{P}_k(\varphi_k)$ to fuse representations of dimension $\mathbb{R}^{d_1+d_2} \rightarrow \mathbb{R}^{d_1}$ in Eq. (5). The paper lacks explicit details regarding its neural architecture, e.g., is it a single linear layer, a multi-layer perceptron, or an attention module?

---

> ### Author Rebuttal · Authors · 2026-03-31
>
> Thank you very much for positive and thoughtful comments. Below, we provide detailed responses to each point.
>
> **W1/Q1**: Thank you for your insightful comments. We agree that robustness to the relative scaling between the local feature dimension $d_1$ and the shared/global feature dimension $d_2$ should be characterized more explicitly. The paper already varies $d_1$ in Fig. 5 and shows that FedARC remains robust across feature dimensions. We further conduct a controlled study over $(d_1,d_2)$.
>
> |$d_2↓\backslash d_1→$|128|256|512|1024|
> |-|-|-|-|-|
> |128|40.09±0.06|43.84±0.09|43.27±0.08|42.21±0.08|
> |256|— |42.86±0.07|**44.21±0.05**|42.65±0.08|
> |512|—|—|43.19±0.08|41.75±0.06|
> |1024|—|—|—|40.78±0.07|
>
> As shown in the additional results, FedARC maintains stable performance across a wide range of $(d_1,d_2)$ combinations, including extreme cases. From a design perspective, this robustness stems from the fact that FedARC does not directly align raw features of different dimensions. Instead, it first projects concatenated representations $[\mathcal{F}^{ex}_k(x),\mathcal{G}^{ex}(x)]\in\mathbb{R}^{d_1+d_2}$ into a unified space via a lightweight projector, and then performs residual compensation (ARC) and semantic alignment (SAA) in this space. This decouples the alignment process from the specific dimensional choices and avoids sensitivity to $d_1$ and $d_2$. We will include these observations and clarify the robustness of FedARC to feature dimension variations in the revised manuscript.
>
> **W2/Q2**: Thank you for your valuable comments. We adopt a systematic hyperparameter tuning strategy based on grid search. The roles of $\lambda$ and $\kappa$ are distinct: $\ell=\ell_{ce}+\lambda · \ell_{mse}$, and $\hat{\mu}\leftarrow(1-\kappa)\hat{\mu}^{t-1}+\kappa\hat{\mu}^{t}.$ Here, $\lambda$ controls the strength of anchor-based semantic regularization, while $\kappa$ controls the bias-variance trade-off of the running semantic statistics. Table 4 already shows a broad stable region, with the best default around $\lambda=1,\kappa=0.1$. Our tuning protocol is: define a coarse grid, tune on a representative cross-device heterogeneous setting, then transfer the best region across datasets/models. Empirically, stronger cross-domain representation shift tends to prefer slightly larger $\lambda$ because the shared anchor must counter larger inter-client semantic drift, while noisier or more unstable settings tend to prefer slightly smaller $\kappa$ because noisier batch means require more conservative running-average updates. We conduct the search over the following space: $\lambda\in ${0,0.01,0.1,1,5,10,20,50,100,200,500} and $\kappa\in ${0,0.1,$\dots$,1.0}. Grid search evaluates combinations over a predefined parameter space to identify configurations with the best performance. Importantly, FedARC does not rely on fragile per-dataset retuning; it works well in a broad neighborhood. We will clarify this procedure in the revised manuscript.
>
> **W3/Q3**: Thank you for pointing this out. In our implementation, the client-specific projector is a single linear layer, $\mathcal{P}_k:\mathbb{R}^{d_1+d_2}\rightarrow \mathbb{R}^{d_1},$ implemented as a single linear layer *nn.Linear(feature\_dim + sub\_feature\_dim, feature\_dim)*. We intentionally avoid using deeper MLPs or attention modules. In FedARC, the projector should remain a lightweight linear interface that exposes mismatches to ARC and SAA. If the projector becomes highly nonlinear, it may absorb part of the cross-client mismatch internally, making the residual vectors less identifiable and weakening compensation. Thus our design deliberately separates roles: projector = linear fusion, ARC = explicit client-wise offset correction, SAA = mean-level cross-client regularization. We verified this empirically:
>
> |Method|single linear layer|2-layer MLP|attention fusion block|
> |-|-|-|-|
> |Acc.|**44.21**|43.83|41.61|
>
> So increasing projector capacity does not improve performance; it degrades it. Nonlinear/attention fusion overfits the fusion step itself, distorts the geometry on which ARC and SAA operate, and makes residual calibration less precise. This motivates our choice of a lightweight linear projector. We will add both the implementation details and this empirical comparison in the revision.

---

> > ### Author Rebuttal · Reviewer_iRz5 · 2026-04-02
> >
> > Thank you for the detailed and thorough rebuttal. The authors have successfully addressed all my concerns, and I will maintain my positive score of 5.

---

### Official Review · Reviewer_QcRZ · 2026-03-12

**Soundness:** 3
**Presentation:** 3
**Significance:** 3
**Originality:** 3
**Overall Recommendation:** 5
**Confidence:** 4

**Summary:**

In realistic deployments, FL is challenged by data and model (architecture diversity) heterogeneity. To address this challenge, the paper introduces FedARC, a federated learning framework that integrates global and local features leveraging learnable projector, while accounting for intra-client and cross-client distribution shifts. Moreover, server-side parameter aggregation is stabilized by incorporating semantic anchor alignment. The experiment results on five datasets show that the proposed method achieves accuracy improvement over various baselines.

**Compliance With Llm Reviewing Policy:**

Affirmed.

**Final Justification:**

The authors have addressed all my concerns and I will keep my positive score of 5.

**Key Questions For Authors:**

None

**Limitations:**

Please see weaknesses and major comments.

**Strengths And Weaknesses:**

Strengths:
- Important and practical problem setting
- Simple but non-trivial integration of various techniques to achieve robust architecture- and data-heterogenous FL framework
- Well presented, easy to follow


Weaknesses:
- Some clarifications and an additional experiment required (see major comments)


Major comments:

1. When it comes to evaluation (inference for final performance reports) in generalization-personalization settings in FL, there are various ways to conduct evaluation (global model with global held-out data, global model with private data in each client, private model with global held-out data, private model with private data) [1]. It seems that FedARC's evaluation is private model with private data. Please, explain the setting for the baselines.

2. Please, provide convergence graph (x-axis: FL rounds, y-axis: accuracy) of FedARC compared to baselines. (Figure 6 in Appendix shows comparison with only FD).

Minor comments:

1. It is not clear what fine-grained feature shifts/batch-level distribution shifts mean throughout Abstract and Introduction. Please elaborate and provide examples.

2. Please add details (both in Introduction and Figure 1 caption) about which specific HtFL method was used in "other HtFLs" in Figure 1c.

3. Section 2.1, line 154 says that only homogeneous feature extractor is updated. Shouldn't homogeneous global headers be aggregated and updated as well? If so, please add. Otherwise, explain the role of homogeneous heads, when each client already possesses private heads.

4. In Table 1, explain N, \rho, \beta briefly in the table caption for self-consistency.


References:

1. Chen, Hong-You, and Wei-Lun Chao. "On Bridging Generic and Personalized Federated Learning for Image Classification." International Conference on Learning Representations.

---

> ### Author Rebuttal · Authors · 2026-03-31
>
> We sincerely thank the reviewer for their time and thoughtful comments. We respond to each point raised in the review and hope our responses address the concerns.
>
> **Major 1**: Thank you for this important question. Our evaluation follows a **private model with private data** protocol, where each client evaluates its own local model $\mathcal{F}_k(\omega_k)$ on its local test set, and the final performance is obtained by averaging across clients. All baselines are trained and evaluated under a unified protocol to ensure fairness. Specifically, each client’s data is split into local training and test sets with a 3:1 ratio, and we report the averaged test accuracy over all clients. We use one local training epoch per round with a batch size of 10, and run each experiment for 1000 communication rounds with a learning rate of 0.01. All baselines are implemented under the same HtFL setting, where clients are assigned heterogeneous models following the HtFE setting (e.g., **HtFE**$^{img}_X$), i.e., different clients use different feature extractors while sharing a homogeneous classifier head. The model assignment is fixed across methods, ensuring that all approaches are evaluated under identical data and model heterogeneity.
>
> Therefore, performance differences are solely attributed to method design rather than discrepancies in evaluation, training, or heterogeneity settings. We will clarify these details in the revised manuscript.
>
> **Major 2**: Thank you for this helpful suggestion. We have included the convergence comparison of FedARC with all baselines and provided the corresponding figure at the following anonymous link: https://anonymous.4open.science/r/FedARC_Figure_6-7562/FedARC_6.png
>
> The results show that FedARC achieves faster convergence and higher accuracy compared to all baselines across communication rounds. We will incorporate this figure into the appendix in the revised version for completeness.
>
> **Minor 1**: Thank you for pointing this out. We agree that these concepts were not sufficiently clear in the current manuscript. In our work, fine-grained feature shifts refer to client-specific deviations in feature representations even within the same class, i.e., for samples $(x_i, y)$ and $(x_j, y)$ from different clients, their embeddings $\mathcal{F}_i(x_i)$ and $\mathcal{F}_j(x_j)$ satisfy $\mathcal{F}_i(x_i) \neq \mathcal{F}_j(x_j)$ due to heterogeneous data and model architectures (e.g., style or background differences). Batch-level distribution shift refers to the variation of representation statistics across local mini-batches, i.e., $\mathbb{E}[\mathcal{F}(x)]$ varies across batches and clients, leading to inconsistent representation alignment during training. We will revise the Abstract and Introduction to include these definitions and illustrative examples for clarity.
>
> **Minor 2**: Thank you for the careful comments. In Figure 1(c), “other HtFLs” actually refers to the best-performing baseline (FD) used in our experiments. We agree that this description may cause confusion. We will revise the figure caption to explicitly specify this for clarity.
>
> **Minor 3**: Thank you for raising this concern. In FedARC, only the homogeneous feature extractor $\mathcal{G}^{ex}$ is aggregated, while the homogeneous classifier heads $\mathcal{G}^{hd}$ are not. This is a deliberate design choice rather than an omission. Specifically, the shared (homogeneous) extractor $\mathcal{G}^{ex}$ is used to establish a common representation space across clients, which is essential for cross-client alignment. Although a global head $\mathcal{G}^{hd}$ exists in the framework, it is not aggregated across clients. Instead, each client maintains its local heterogeneous classifier head $\mathcal{F}_k^{hd}$ for prediction, while the global branch facilitates representation alignment (e.g., via SAA) during training. Aggregating classifier heads $\mathcal{G}^{hd}$ would introduce additional instability under heterogeneous data and models, and is unnecessary for aligning representations. FedARC focuses on representation-level consistency rather than parameter sharing at the classifier level. We will clarify this design choice and the role of the global head in the revised manuscript to avoid confusion.
>
> **Minor 4**: Thank you for this constructive suggestion. In Table 1, $N$ denotes the total number of clients, $\rho$ represents the client participation ratio, and $\beta$ is the Dirichlet concentration parameter controlling the degree of label skew. We will add these explanations to the table caption for improved clarity and readability in the revised manuscript.

---

> > ### Author Rebuttal · Reviewer_QcRZ · 2026-04-02
> >
> > Thank you for your detailed and careful rebuttal.
> >
> > You have addressed all my concerns and I believe adding minor clarifications (for readability) along with the convergence graphs either into the main text or the Appendix will be helpful for overall comprehensiveness of the work.
> >
> > Thanks again and I will keep my positive score 5.

---

### Official Review · Reviewer_5W1i · 2026-03-13

**Soundness:** 2
**Presentation:** 2
**Significance:** 2
**Originality:** 2
**Overall Recommendation:** 4
**Confidence:** 3

**Summary:**

This paper studies heterogeneous federated learning. In particular, the authors motivate their problem from the perspective of **feature drift** across clients, where local feature distributions differ and lead to representation misalignment during federated training. To address this issue, the paper proposes FedARC, which introduces a framework combining shared global parameters and client-specific parameters with an adaptive residual compensation mechanism. The approach also incorporates a projection module and semantic anchor alignment to mitigate representation discrepancies between clients. Empirical experiments on five datasets under **label shift** federated learning settings are provided, together with a convergence analysis under non-convex objectives.

**Compliance With Llm Reviewing Policy:**

Affirmed.

**Final Justification:**

All my questions are addressed. The authors have made a huge effort in comparing with baselines during rebuttal.

**Key Questions For Authors:**

- The paper motivates the method using feature drift across clients, but the experimental evaluation appears to focus primarily on label shift settings. Can the authors provide experimental results under explicit feature shift scenarios to validate the motivation of the method?
- The reviewer is not fully convinced the necessesity of studying model heterogeneity in practice. Could the authors elaborate on the practical scenarios where model heterogeneity plays a critical role?
- The proposed approach combines shared parameters and client-specific parameters (in a different way) for personalization. Similar ideas have been explored in several prior works (e.g., FedPer, pFedMe, FedRep, Ditto). The authors briefly discuss these works in the related work section, but the paper would be stronger if the authors could provide a concrete example explaining the underlying mechanism that makes the proposed method more effective. Otherwise, the observed performance gains could potentially be attributed to the more complex model architecture.
- What is the computational and communication complexity of the system as the number of clients $N$ increases? Are there any potential scalability issues?

**Limitations:**

yes

**Strengths And Weaknesses:**

**Strengths**

- The method and the algorithmic pipeline are clearly described.
- The paper provides some theoretical convergence analysis.

**Weaknesses**

- **Mismatch between motivation and experimental evaluation:** The paper motivates the proposed method primarily from the perspective of **feature drift across clients**, suggesting that the main challenge lies in differences in feature distributions across local datasets. However, the experimental settings appear to focus primarily on **label shift** (e.g., Dirichlet-based partitioning), which mainly alters the label distribution $P(y)$ rather than the feature distribution $P(x)$. As a result, the empirical evaluation may not adequately test the core problem the method is intended to address. It would strengthen the paper to include experiments that explicitly simulate feature distribution shifts across clients, such as domain-shifted datasets or feature-partitioned benchmarks. Works in [1-3] has such experiment settings.
- **Empirical results are not extensive**: The experiment setting only considers mild labels shift ($\beta=0.1$). For a paper that highlights heterogeneity as one of its key motivations, more extensive experiments would be desirable. In particular, the evaluation could include different types of heterogeneity and varying levels of severity.
- **Algorithmic complexity:** The proposed framework introduces several interacting components, including projection modules, residual compensation, semantic anchor alignment, and shared/personalized parameters. While each component is motivated individually, it is not entirely clear that all of them are necessary to achieve the reported performance improvements. The resulting method appears relatively complicated compared to existing personalization-based FL approaches. Without a clearer minimal formulation or stronger ablation studies demonstrating the necessity of each component, the overall design risks appearing overly engineered.
- **Theoretical results are somewhat standard:** While the paper provides convergence guarantees under non-convex objectives, the rate $O(1/T)$ is standard in federated optimization analysis and does not appear to provide deeper insights into the behavior of the proposed algorithm. It is therefore unclear how strongly the theoretical analysis supports the practical advantages of the method.

References:

1. Beyond Aggregation: Guiding Clients in Heterogeneous Federated Learning
2. Is Heterogeneity Notorious? Taming Heterogeneity to
Handle Test-Time Shift in Federated Learning
3. FedBN: Federated learning on non-iid features via local batch normalization

---

> ### Author Rebuttal · Authors · 2026-03-31
>
> **W1/Q1**: Thank you for highlighting this important point, which is indeed critical for validating the motivation under feature shift. Actually, our appendix already includes feature-shift experiments on DomainNet (Tables 10–11). Each client corresponds to a distinct domain, where clients have an identical number of labels but differ in the features of their data [1]. The accompanying ablation results further demonstrate that ARC and SAA effectively address representation inconsistency caused by feature shift. further evidence, we report the performance below (std omitted due to space). We will clarify this setting in the revision.
>
> |Method|Clipart|Infograph|Painting|Quickdraw|Real|Sketch|Avg|
> |-|-|-|-|-|-|-|-|
> |FD|26.05|17.01|24.16|59.86|31.70|21.40|30.03|
> |FedProto|19.22|13.79|19.76|55.11|22.25|16.80|24.49|
> |FedTGP|21.82|15.88|20.24|59.20|23.70|18.40|26.54|
> |LG-FedAvg|27.69|16.54|24.07|56.03|28.12|21.05|28.91|
> |FedGen|27.62|15.49|23.82|56.09|28.92|19.76|28.62|
> |FedGH|25.52|16.15|22.77|56.66|27.50|19.20|27.97|
> |pFedES|25.91|17.07|24.20|59.90|29.74|21.44|29.71|
> |FedKD|25.42|17.61|23.59|58.64|29.41|21.56|29.37|
> |FedMRL|25.36|16.95|23.32|58.24|29.65|21.10|29.10|
> |FedARC|30.54|20.13|25.50|61.62|32.63|29.28|33.28|
>
> [1] HtFLlib: A Comprehensive Heterogeneous Federated Learning Library and Benchmark. ACM SIGKDD. 2025.
>
> **W2/Q2**: Thank you for this valuable suggestion. We would like to further clarify that our submission already covers a broad range of heterogeneous settings. Specifically, we report results (Table 8) under multiple Dirichlet label-skew levels ($\beta$ = 0.01, 0.1, 0.5, 1). We further include more challenging non-IID scenarios via the Pathological Setting [1][2] (data partition shown in Fig. 13) and additional experiments in Table 12.
>
> Beyond data heterogeneity, we also evaluate model heterogeneity (Table 2) by considering different levels of architectural inconsistency across clients [1]. This is important, as model heterogeneity is a key challenge in HtFL [1][3][4]. From a practical perspective, real-world federated systems inherently involve both data and model heterogeneity due to diverse user behaviors and hardware constraints. Evaluating under such diverse settings is essential to demonstrate the robustness and applicability of our method.
>
> Additional results show FedARC still achieves the best accuracy in ($\beta=0.05$):
>
> |Method|FD|FedTGP|LG-FedAvg|FedGen|FedGH|pFedES|FedKD|FedMRL|**Ours**|
> |-|-|-|-|-|-|-|-|-|-|
> |Acc.(Rnd)|56.74(181)|55.81(308)|54.22(192)|52.10(165)|53.97(233)|55.83(210)|56.21(189)|55.46(173)|**59.49 (155)**|
>
> [2] One Arrow, Two Hawks: Sharpness-aware Minimization for Federated Learning via Global Model Trajectory. ICML 2025.
>
> [3] Bridging Generalization Gap of Heterogeneous Federated Clients Using Generative Models. ICLR 2026.
>
> [4] Beyond Aggregation: Guiding Clients in Heterogeneous Federated Learning. ICLR 2026.
>
> **W3/Q3**: Thank you for this comment. Actually, FedARC is fundamentally different from methods such as FedPer, pFedMe, FedRep, and Ditto. These methods primarily rely on parameter-level personalization, without explicitly modeling feature discrepancies across clients. In contrast, FedARC operates in the representation space: it fuses local and global features, uses ARC to compensate client-specific feature deviations, and employs SAA to enforce cross-client semantic alignment. The key difference is that FedARC directly models and corrects feature misalignment in the representation space, whereas the mentioned methods do not explicitly address such discrepancies. This is supported by Tables 10–11, where removing ARC or SAA consistently degrades performance, demonstrating that each component is necessary and tightly coupled in addressing feature shift.
>
> **W4**: We agree that the $\mathcal{O}(1/T)$ convergence rate is standard in FL optimization. Our focus is to provide theoretical guarantees for convergence under heterogeneous settings, rather than deriving a new convergence rate. This design choice is consistent with recent works such as pFedES (AAAI 2025) and FedClean (ICML 2025), which also adopt standard convergence rates while emphasizing stability and robustness under non-IID data. The key role of our analysis is to justify the soundness and convergence of FedARC under data and model heterogeneity, rather than to optimize the rate itself.
>
> **Q4**: Thank you for this insightful question. For communication, FedARC merely communicates parts of  small homogeneous model. Thus, the additional communication overhead is relatively modest. For computation,  FedARC is comparable to standard HtFL training. The additional cost is lightweight and operate on low-dimensional representations, making it negligible compared to backbone model training. Importantly, FedARC does not require additional client-to-client communication or extra synchronization steps, thus ensuring scalability with the number of clients

---

> > ### Author Rebuttal · Reviewer_5W1i · 2026-04-03
> >
> > Thanks for the rebuttal. For the experiment in Q1, can you compare it against methods such as FedBN, FedBABU, and FedDRM? I am curious about the performance. Please also clarify the experiment setting that you use when comparing against these methods.

---

> > > ### Author Response · Authors · 2026-04-06
> > >
> > > Thank you for the helpful suggestion. We have added comparisons with FedBN, FedBABU, and FedDRM under the same HtFL benchmark as FedARC. We also clarify that FedBN and FedBABU were originally proposed for homogeneous FL / personalization settings, rather than native fully heterogeneous FL. In contrast, recent studies have focused on the more realistic federated setting with both data and model heterogeneity, which is also the main focus of this work [1,2,3,4,5]. Therefore, when adapting FedBN and FedBABU to the HtFL framework, we preserved their original design principles as faithfully as possible.
> > >
> > > **Cross-silo (N=10, ρ=100%, β=0.1)**
> > > |Method|Cifar10|Cifar100|Flower102|Tiny*|
> > > |-|-|-|-|-|
> > > |FedBN|84.37±0.16|**44.67±0.04**|51.33±0.21|27.29±0.26|
> > > |FedBABU| 86.42±0.12|44.09±0.07|51.81±0.19|26.67±0.15|
> > > |FedDRM|71.71±0.17|31.44±0.02|42.05±0.15| 20.10±0.18|
> > > |FedARC|**89.22±0.02**|44.21±0.05|**57.67±0.11**|**29.91±0.14**|
> > >
> > > **Cross-device (N=50, ρ=20%, β=0.1)**
> > > |Method|Cifar10|Cifar100|Flower102|Tiny*|
> > > |-|-|-|-|-|
> > > |FedBN|83.16±0.20|40.07±0.10|45.33±0.28|23.89±0.21|
> > > |FedBABU|83.53±0.18|39.91±0.11|44.33±0.22|23.72±0.18|
> > > |FedDRM|70.06±0.21|30.13±0.23|40.33±0.23|18.01±0.20|
> > > |FedARC|**86.14±0.14**|**42.25±0.07**|**50.18±0.15**|**27.10±0.16**|
> > >
> > > The results show that FedBN remains highly competitive on Cifar100, where keeping BN local effectively absorbs client-specific feature statistics. However, under the more challenging cross-device setting with partial participation, its advantage diminishes because local BN mainly corrects local statistics, while cross-client semantic drift and representation misalignment remain unaddressed. FedBABU also performs strongly, since its frozen head / body learning strategy encourages stronger shared representations before personalization, which is beneficial for more complex multi-class tasks. However, this comes at a much higher computation cost—typically several times that of other HtFL methods—which makes it less suitable for realistic edge scenarios with limited resources.
> > >
> > > By contrast, FedDRM is substantially less robust in the benchmark. Although it is an HtFL method, its official implementation is tuned for a rather different setting (e.g., 10 local steps, and batch size 128). Its routing/client-classification branch relies on sufficiently stable mini-batch statistics and reliable client-density estimation. Under the unified HtFL benchmark, where local batches are much smaller and the setting is more heterogeneous and sample-scarce, this routing branch becomes much harder to optimize, leading to a marked drop in overall accuracy.
> > >
> > > All methods were evaluated under the unified HtFL setting used in the paper, following the HtFLlib benchmark [1]: HtFE$^{img}$$_8$ heterogeneous models, label-skew partition with $\beta$=0.1, cross-silo with $N$=10, $\rho$=100%, cross-device with $N$=50, $\rho$=20%, one local epoch per selected client, batch size 10, learning rate 0.01, unified feature dimension 512, and 1000 communication rounds. Within this common setting, we kept each baseline’s core mechanism unchanged: FedBN uses local BN with non-BN aggregation, FedBABU uses frozen-head/body-only training with personalization-time fine-tuning, and FedDRM keeps its dual-loss and routing design while being adapted to the HtFLlib framework[1] . FedARC remains the most consistently strong method across datasets and settings, while FedBN only achieves a dataset-specific peak on cross-silo Cifar100. We will add these results and clarifications to the revised manuscript.
> > >
> > > [1] HtFLlib: A Comprehensive Heterogeneous Federated Learning Library and Benchmark. ACM SIGKDD. 2025.
> > >
> > > [2] FedSA: A Unified Representation Learning via Semantic Anchors for Prototype-based Federated Learning. AAAI 2025.
> > >
> > > [3] pFedES: Generalized Proxy Feature Extractor Sharing for Model Heterogeneous Personalized Federated Learning. AAAI.  2025.
> > >
> > > [4] Federated model heterogeneous matryoshka representation learning. NeurIPS Proceedings.
> > >
> > > [5] FedTGP: Trainable Global Prototypes with Adaptive-Margin-Enhanced Contrastive Learning for Data and Model Heterogeneity in Federated Learning. AAAI.

---

### Decision · Program_Chairs · 2026-04-30

**Decision:**

Accept (spotlight)

**Comment:**

This paper proposes FedARC, a heterogeneous federated learning framework that mitigates feature drift across clients via shared global parameters, client-specific projectors, adaptive residual compensation, and semantic anchor alignment.

Reviewer 5W1i: The motivation focuses on feature drift but experiments only simulate label shift; the evaluation lacks diversity in heterogeneity types and severity, the method appears over-engineered without strong ablation support, and the convergence analysis offers standard rates without deep algorithmic insight.

Reviewer QcRZ: The evaluation protocol for personalization is not clearly specified for baselines, and a convergence graph (rounds vs. accuracy) comparing all methods is missing, along with several minor clarifications on terminology, figure details, and table captions.

Reviewer iRz5: The robustness to varying feature dimensions (local vs. global) is not fully characterized, hyperparameter selection procedures are unspecified, and the architecture of the client‑specific projector lacks explicit detail.

Reviewer Zu6U: The global semantic anchor is fixed after initialization and not updated adaptively, scalability to hundreds or thousands of clients is not demonstrated, privacy risks of sharing semantic statistics are not discussed, and the client–server communication of semantic anchors is unclear.

Overall conclusion: The authors have comprehensively addressed all reviewer concerns through additional experiments, ablation studies, clarifications, and theoretical discussions; therefore, the paper is recommended for acceptance.